



# Do Southern Hemisphere tree rings record past volcanic events?

Philippa A. Higgins[1,2], Jonathan G. Palmer[2,3], Chris S. M. Turney[2,3], Martin S. Andersen[4], Fiona Johnson[1]

[1]Water Research Centre, School of Civil and Environmental Engineering, UNSW Sydney, NSW, 2052, Australia
[2]ARC Centre of Excellence for Australian Biodiversity and Heritage, University of New South Wales, NSW 2052, Australia
[3]Earth and Sustainability Science Research Centre, School of Biological, Earth and Environmental Sciences, University of New South Wales, NSW 2052, Australia
[4]Water Research Laboratory, School of Civil & Environmental Engineering, UNSW Sydney, NSW 2052, Australia

*Correspondence to*: Philippa A. Higgins (Philippa.higgins@unsw.edu.au)

## Abstract

Much of our knowledge about the impacts of volcanic events on climate comes from proxy records. However, little is known about the impact of volcanoes on trees from the Southern Hemisphere. We investigated whether volcanic signals could be identified in ring widths from eight New Zealand dendrochronological species, using superposed epoch analysis. We found that most species are good recorders of volcanic dimming and that the magnitude and persistence of the post-event response can be broadly linked to plant life history traits. Across species, site-based factors, particularly altitude and exposure to prevailing conditions, are more important determinants of the strength of the volcanic response than the species. We then investigated whether proxy selection impacts the magnitude of post-volcanic cooling in tree-ring based temperature reconstructions by developing two new multispecies reconstructions of New Zealand summer (December-February) temperature. Both reconstructions showed temperature anomalies remarkably consistent with studies based on instrumental temperature, and with the ensemble mean response of climate models, demonstrating that New Zealand ring widths are reliable indicators of regional volcanic climate response. However, we also found that volcanic response is complex, with positive, negative, and neutral responses identified - sometimes within the same species group. Species-wide composites thus tend to underestimate the volcanic response. The has important implications for the development of future tree ring and multiproxy temperature reconstructions from the Southern Hemisphere.

## 1 Introduction

Emissions from large volcanic eruptions are a key source of temperature and hydroclimate variability on interannual to decadal time scales (Iles et al., 2015; Robock, 2000). As few large volcanic eruptions have occurred during the instrumental era, much of our knowledge about volcanic impacts on climate, particularly regional and global temperature, comes from proxy records (Tejedor et al., 2021). These records are predominantly high altitude or high latitude tree-ring proxies from the Northern Hemisphere (e.g. Briffa et al., 1998; D'Arrigo et al., 2009; Pieper et al., 2014). In comparison, there are very few proxy-based characterisations of the temperature response to volcanic events from the Southern Hemisphere (Neukom et al., 2014; Tejedor



et al., 2021). The few studies considering Southern Hemisphere tree-ring proxies have not found significant impacts in the years following large volcanic eruptions (Allen et al., 2018; Cook et al., 1992; Krakauer & Randerson, 2003; Palmer & Ogden, 1992).

Whether the hemispheres have contrasting sensitivities to volcanic eruptions is vital to understanding future climate projections (Neukom et al., 2014). The muted volcanic impact in Southern Hemisphere proxy reconstructions could be due to a maritime dampening effect on post-eruption cooling and/or the distribution of landmasses toward the Equator (Allen et al., 2018; Krakauer & Randerson, 2003; Raible et al., 2016). Such explanations suggest the magnitude of Southern Hemisphere cooling is too small to be reliably recorded in tree-ring archives. However, climate models show a clear Southern Hemisphere volcanic

signal (Neukom et al., 2014, 2018). The reasons for the considerable discrepancy between proxy reconstructions and climate models in the Southern Hemisphere remain unclear. Potential explanations for the model/data discrepancy are changes in the hydrological cycle in response to volcanic cooling, with the models underestimating ocean moderation of reduced post-eruption cooling, uncertainties in volcanic forcing data, and/or proxy noise and spatial distribution (Neukom et al., 2018; Zhu et al., 2020).

The question remains as to whether Southern Hemisphere proxies – specifically tree rings - do record volcanic events. To our knowledge, no studies have explored the factors which influence whether (or not) volcanic signals can be identified using tree ring data from the Southern Hemisphere. Tree growth is dependent on a range of environmental and biological factors, so careful site and tree selection is necessary to ensure that a specific influence of interest can be studied (Norton & Ogden, 1987). Northern Hemisphere tree-ring studies are predominantly from high-latitudes or mid-latitude alpine timberline sites, where

tree growth is temperature limited (Scuderi, 1990). Around 80% of chronologies from such sites show significant growth reductions following large eruptions due to unusually low growing-season temperatures (Krakauer & Randerson, 2003). Tree-ring studies from Northern Hemisphere mid-latitude lowland sites have shown that volcanic response is less clear, as temperate zone trees are less temperature-limited and have more complex relationships to multiple climate variables (Pieper et al., 2014; Wilson et al., 2016).

Exploring possible responses to volcanic eruptions, Pieper et al. (2014) proposed three models for tree growth in temperate regions: 1) Growth reduction due to volcanic cooling, resulting in narrow rings; 2) Neutral/ no response if the climate sensitivity to volcanic eruptions is insufficient to influence tree growth; or 3) Enhanced growth due to an increase in the diffuse light fraction and reduced water stress, resulting in wide rings. Temperate-zone trees from the Southern Hemisphere are also likely to display similar mixed volcanic signals, depending on their relative sensitivity to growing season temperatures and

water stress and the magnitude of the regional cooling effect. Understanding these factors will help elucidate the proxy archive contribution to the Southern Hemisphere model-data discrepancy. This knowledge will benefit future studies of hemispheric temperatures and help identify which species and/or regions should be prioritised for future proxy development.

Our goal in this study is to assess whether Southern Hemisphere tree rings record past volcanic events using a multispecies network of high-quality, replicated tree-ring chronologies from New Zealand. This country is a long, narrow, archipelagic

landscape stretching from 34°S to 47°S. Climatically, the northern part protrudes into the warm sub-tropical ridge, whereas



the southern end is embedded in the cool southwesterlies (Salinger, 1980). The North and South Island axial ranges, which rise to 3,764 m, are a significant barrier to east-west airflow, leading to strong regionalisation of precipitation anomalies (Salinger, 1980). Land clearing after European settlement resulted in the loss of most lowland forests and nearly all forests from the eastern drylands of New Zealand, with remaining wet conifer-broadleaved forests and montane to alpine southern

beeches (*Nothofagaceae*) dominant forests now most common (McGlone et al., 2017).

Tree-ring chronologies have been developed from locations widely distributed throughout New Zealand. Since the initial dendrochronological studies were undertaken by LaMarche et al. (1979), records have been generated from nine endemic species, of which seven are conifers, and two are *Nothofagaceae* (Table 1). Five main species have been used to develop multi-centennial tree-ring chronologies: kauri (*Agathis australis*), pink pine (*Halocarpus biformis*), silver pine (*Manoao colensoi*),

cedar (*Libocedrus bidwillii*), and silver beech (*Lophozonia menziesii*). Most chronologies are primarily sensitive to austral summer temperatures, with temperature reconstructions developed from beech (Norton et al., 1989), silver pine (Cook et al., 2002), cedar (Palmer & Xiong, 2004), pink pine (Duncan et al., 2010), and multispecies networks (Salinger et al., 1994). Thus, New Zealand, with its wide latitudinal and altitudinal range, regionalised climate zones, and wide distribution of tree-ring chronologies from multiple species – including multiple species from the same site - is ideal for studying tree-ring sensitivities

to past volcanic events. Using the New Zealand dendrochronological dataset, we aim to answer the following specific questions:

1. Can we identify volcanic signals in the tree-ring series?
2. Are there differences in the expression of volcanic signals between the species?
3. Does proxy selection impact the magnitude of post-volcanic cooling in tree-ring based temperature reconstructions?

## 90 2 Data and methods

### 2.1 Tree-ring chronologies

The New Zealand tree-ring chronologies analysed in this study were collated to develop the Eastern Australia and New Zealand Drought Atlas (Palmer et al., 2015; Fig. 1). Palmer et al. (2015) identified chronologies from the International Tree Ring Data Bank and personal collections, screened the tree-ring measurements for dating problems using the software program

COFECHA (Holmes 1983, Grissino-Mayer 2001), and developed site 'master' chronologies from the raw ring widths using the 'signal free' method of standardisation (Melvin & Briffa, 2008). The metadata for all chronologies is listed in the supplement to Palmer et al. (2015). We excluded the single chronology developed from mountain toatoa (*Phyllocladus alpinus*) from this list due to insufficient species depth, leaving 96 chronologies from 8 dendrochronological species for volcanic response analysis.

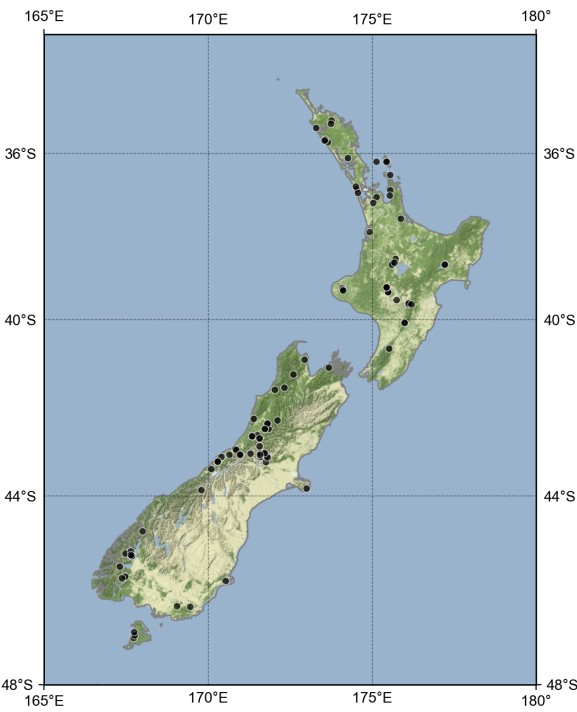

**Figure 1 – Distribution of tree-ring chronologies in New Zealand. Made with Natural Earth free vector and raster map data @ naturalearthdata.com.**

## 2.2 Selection of volcanic events

A significant source of uncertainty in tree-ring studies of volcanic signals is event selection. The choice of volcanic events can greatly influence the magnitude of average regional cooling identified (Esper et al., 2013; Wilson et al., 2016). In addition, for many events which occurred before instrumental records, the timing, location, and size of eruptions are uncertain (Garrison et al., 2018; Timmreck et al., 2021). For this analysis, we are interested in those events which would likely have reduced growing season temperatures over New Zealand and thus be identifiable as ring-width anomalies. Therefore, we selected events using a regional volcanic dimming threshold rather than an eruption magnitude. Prior to the instrumental era, we picked events from the Greenland and Antarctic ice core sulphate aerosol analysis of Toohey & Sigl (2017) based on peak stratospheric atmospheric aerosol depth (SAOD). We averaged SAOD, modelled using the Easy Volcanic Aerosol module (Toohey et al., 2016) over the latitudinal range of New Zealand (30 to 50° S) and selected two thresholds: SAOD > 0.08, which resulted in 10 eruptions for analysis between 1400 and 1900 CE; and SAOD > 0.04 resulting in 18 eruptions for the same period (Figure 2). Two thresholds were chosen as we cannot know, prior to analysis, what SAOD magnitude corresponds to a substantial temperature response, yet selecting an event magnitude post-analysis risks biasing the results (Haurwitz & Brier, 1981). Between 1900 and 1990, we selected the three largest tropical eruptions, which have been shown to have significant impacts on instrumental temperatures in New Zealand (Salinger, 1998). At each methodological step, the analysis was carried out using both the 13 and 21 event lists separately. Full details of all selected eruptions are provided in Table S1.





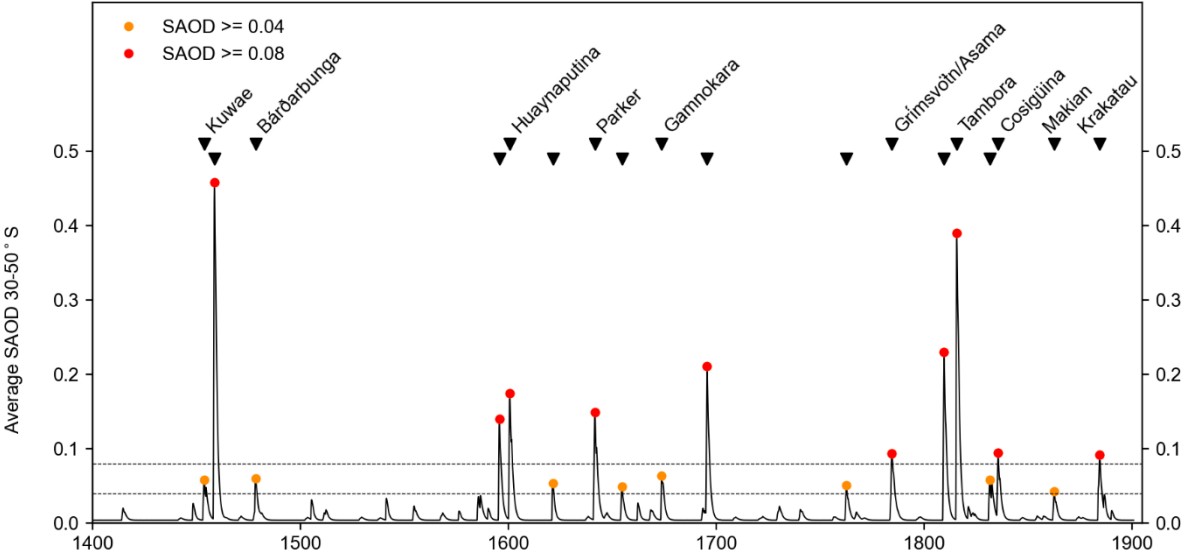

**Figure 2 – Selection of volcanic events based on thresholds of peak modelled stratospheric atmospheric optical depth (Toohey & Sigl, 2017), averaged over 30-50 ºS, greater than 0.04 (orange) and 0.08 (red). Known eruptions are labelled.**

### 2.3 Superposed epoch analysis

We tested whether a volcanic signal can be identified in New Zealand tree-ring chronologies using superposed epoch analysis (SEA; (Haurwitz & Brier, 1981)), a statistical technique widely used to identify the impacts of volcanic eruptions on climate (Adams et al., 2003; Rao et al., 2019; Salinger, 1998; Scuderi, 1990; Tejedor et al., 2021). The composite response of individual site master chronologies to the 13 (SAOD > 0.08) or 21 (SAOD > 0.04) volcanic events between 1400 and 1990 CE was studied 0 - 5 years post-event, with anomalies calculated by subtracting the average of the nearest 5-year background period undisturbed by volcanic forcing (Table S1; (Büntgen et al., 2020)). Species-level responses were then tested using a composite chronology produced by simple averaging of annual values across sites (Cook & Kairiukstis, 1990). Volcanic responses were categorised as positive or negative if the anomalies exceeded the 5th or 95th percentile response of 10,000 random samples of years undisturbed by volcanic forcing, or neutral if they fell between these bounds.

To directly compare the volcanic sensitivities of species, we selected six sites where chronologies have been developed from two species. For these sites, we compared the SEA results carried out on the individual, standardised, tree-ring series contributing to the site master chronology, provided the length of the series allowed for at least eight volcanic events to be analysed. Only the 21-event list was used for analysis as too many series were dropped using the more restricted 13-event set. The difference in response, calculated as the sum of the normalised five-year post-event anomaly across all contributing series and events, were compared using the non-parametric Mann-Whitney U-test.



## 2.4 Temperature reconstructions

To investigate the influence of proxy selection on the identification of volcanic signals in temperature reconstructions, two
new reconstructions of New Zealand average temperatures are reported here. We used the New Zealand average 'seven-station'
monthly instrumental temperature series (Mullan, 2012; Salinger, 1981), obtained from the New Zealand National Institute of
Water and Atmospheric Research (NIWA), to examine the temperature response of the chronologies. To ensure sufficient
overlap between the chronologies and temperature dataset, only chronologies extending to or beyond 1990 were retained for
analysis. As many sites have not been updated since they were originally sampled in the 1970/80s, only 58 of the 96 available
chronologies were included in the temperature analysis. Correlations were calculated between autoregressively modelled
chronologies and monthly climate data from 1911-1990, with each month treated as a separate time series. A 20-month window
was selected for correlation analysis, extending from October of the previous growing season to May at the end of the current
austral growing season. Two growing seasons were included as significant prior season climate sensitivities have been reported
for some species (Table 1). Based on the response analysis, December-February (DJF) was selected as the seasonal target, as
this window captures the strongest correlations across all species (see Results).

The first reconstruction (NZall) included the full suite of available chronologies extending to 1990, while the second (NZsel)
was limited to those chronologies which showed a significant volcanic signal using SEA. In each case, only those chronologies
significantly ($p < 0.1$) correlated to average DJF temperatures over the period 1911-1990 were used. Tree ring series were also
tested as potential predictors with a lag of one year with respect to the temperature data, as prior year climate often has a
lingering influence on current year tree growth (Fritts, 1976). Average DJF temperatures were reconstructed using nested
principal components linear regression (Cook et al., 1999, 2007, 2010). A 50:50 split calibration-validation scheme was used,
in which the model was initially calibrated on the first half of the data (1911-1950) and validated on the second half (1951-
1990), then the model was re-estimated with the calibration and validation periods reversed. Once the split models were
verified based on the verification period reduction of error (VRE) and verification period coefficient of efficiency (VCE; (Cook
& Kairiukstis, 1990)) metrics, the entire data period was used to produce the final reconstructions (Briffa et al., 1990).

The volcanic response in tree-ring reconstructions of temperature was also tested using SEA and the two sets of volcanic
eruption years. Further, variation in the temperature response to different volcanic events was estimated by calculating the
90th percentile bootstrap confidence interval from 1,000 replicates drawn without replacement from the event list (Rao et al.,
2019). In each iteration, approximately two-thirds (9/13 or 15/21) of the volcanic events were selected. The confidence interval
provides some indication of how eruptions of different sizes, locations, and seasonality may impact the SEA results. We
compared the volcanic response seen in our multispecies reconstructions to the ensemble mean response of seven climate
models from the Coupled Model Intercomparison Project 5 (CMIP5) suite with Last Millennium (past1000, 850-1850 CE)
simulations. The CMIP5 models were forced with either the Gao et al. (2008) or Crowley and Unterman (2013) volcanic
forcing series (see Table S2). Data from the historical simulations were appended to extend the dataset from1850 to 2005.




**Table 1 – Distribution, reported climate sensitivities and key references for New Zealand dendrochronological species**

| Code | Species | Common name | No. chronologies | Ring width (mm/y) | Persistence (GINI²) | Distribution | Reported climate sensitivity | Chronology development |
|---|---|---|---|---|---|---|---|---|
| AGAU | *Agathis australis* | Kauri | 17 (9*)¹ | 1.66 ± 0.59 | 0.106 | North Island, north of 38°S; predominantly lowland forests, can be > 500 m | ENSO; inverse relationship to current year temperature and precipitation. | (Boswijk et al., 2006; Buckley et al., 2000; Fowler et al., 2008; LaMarche et al., 1979; Ogden & Ahmed, 1989; Palmer et al., 2006) |
| HABI | *Halocarpus biformis* | Pink pine | 20 (19*) | 0.44 ± 0.1 | 0.074 | Low altitude to sub-alpine; central North Island to Stewart Island | Frost tolerant; sensitive to year-round temperatures | (R. D. D'Arrigo et al., 1996; Fenwick, 2003; Xiong et al., 1998) |
| LACO | *Manoao colensoi* (formerly *Lagerostrobos colensoi*) | Silver pine | 6 (4) | 0.56 ± 0.20 | 0.065 | Low-elevation forests of the South Island west coast and some North Island locations. | Summer temperatures | (Cook et al., 2002; R. D. D'Arrigo et al., 1998) |
| LIBI | *Libocedrus bidwillii* | New Zealand cedar | 26 (21) | 0.7 ± 0.17 | 0.091 | Widely distributed over North and South Islands south of 38°S; 200 to 1200 m above sea level | Summer temperatures, precipitation | (LaMarche et al., 1979; Xiong & Palmer, 2000) |
| NOME | *Lophozonia menziesii* (formerly *Nothofagus menziesii*) | Silver beech | 7 (1) | 1.14 ± 0.24 | 0.136 | Montane and subalpine forests, common in the western South Island | Summer temperatures | (Norton, 1983b, 1984) |
| NOSO | *Fuscospora cliffortioides* (formerly *Nothofagus solandri* var. *cliffortioides*) | Mountain beech | 11 (4) | 0.92 ± 0.23 | 0.136 | Closed forests of the central North Island and the eastern South Island, valley floor to ~1300 m | Summer temperatures | (Norton, 1983a, 1984) |
| PHAL | *Phyllocladus alpinus* | Mountain toatoa | 1 | 0.59 ± 0.22 | 0.065 | Throughout New Zealand, lowland to subalpine forest | N/A | (LaMarche et al., 1979) |
| PHGL | *Phyllocladus toatoa* (formerly *P. glaucus*) | Toatoa | 4 | 0.63 | 0.129 | North Island, montane forest between 850 and 1000 m | Summer temperatures, precipitation, pressure anomalies | (Dunwiddie, 1979; LaMarche et al., 1979; Palmer, 1989; Salinger et al., 1994) |
| PHTR | *Phyllocladus trichomanoides* | Tanekaha | 5 | 1.04 | 0.118 | Lowland forest up to 800 m above sea level, north of 42°S | Summer temperatures, precipitation, pressure anomalies | (Dunwiddie, 1979; LaMarche et al., 1979; Palmer, 1989; Palmer & Ogden, 1992; Salinger et al., 1994) |

¹ Number in brackets indicate the number of chronologies extending to 1990 available for the temperature reconstructions. *Includes master chronology.
² GINI coefficient – an all-lag measure of diversity in tree-ring records (Biondi & Qeadan, 2008)




## 3. Results

### 3.1 Overall species volcanic responses

Eight tree species were analysed for a volcanic response using superposed epoch analysis. Figure 3 shows the response of SEA based on the 13 volcanic eruptions with SAOD > 0.08 between 1400 and 1990 CE. Figure 3 shows two composite responses;
the species-wide composite, averaged across all sites, and a 'sensitive chronology' composite, calculated from the site chronologies, which individually showed either a significant positive or significant negative response to volcanic eruptions. The species-wide response to volcanic events varied widely between New Zealand dendrochronological species. Three out of eight species recorded a composite neutral response. Of the remaining five species, one recorded a positive response and four recorded a negative response. Analysis was repeated for the 21 eruptions with SAOD > 0.04 with similar, but weaker, results
for most species, suggesting not all events had a measurable climatic impact over New Zealand (Fig. S1).

Kauri (Fig. 3 a) was the only species to show a composite positive response to volcanic events, maximal in year t+1. Kauri showed a consistent response across sites with all except two chronologies showing a positive anomaly following an event, although only 8/17 positive responses were significant at $p < 0.05$. The sensitive chronology composite recorded a very strong t+1 response, indicating that at these eight sites, kauri receives a significant growth benefit from the climatic changes following
a volcanic eruption.

Pink pine, cedar, silver beech, and toatoa show lagged negative responses to volcanic events, with peak negative anomalies recorded in years t+2 or t+3 (Fig.3 b, c, e, g). The pink pine and toatoa responses are broadly consistent within their respective groups. All except one pink pine chronology recorded a negative response in t+2, which was significant for most of the chronologies (14/21). All four toatoa chronologies recorded a significant negative response in t+3. In contrast, cedar does not
show a consistent species-wide response; both significant negative and positive responses were recorded in 13/26 chronologies, with the remainder showing a neutral response. The cedar chronologies have the widest geographical distribution of any species, and geographical factors that may influence the variability in response are considered in Section 3.3 below.

Minimal volcanic response was recorded by the two *Nothofagacae* species composites, although individual chronologies did record a response (Fig 3. e, f). As many beech chronologies extend only to the mid-1700s, the species composites were tested
against a smaller subset of volcanic events, which may contribute to the subdued species-wide responses.



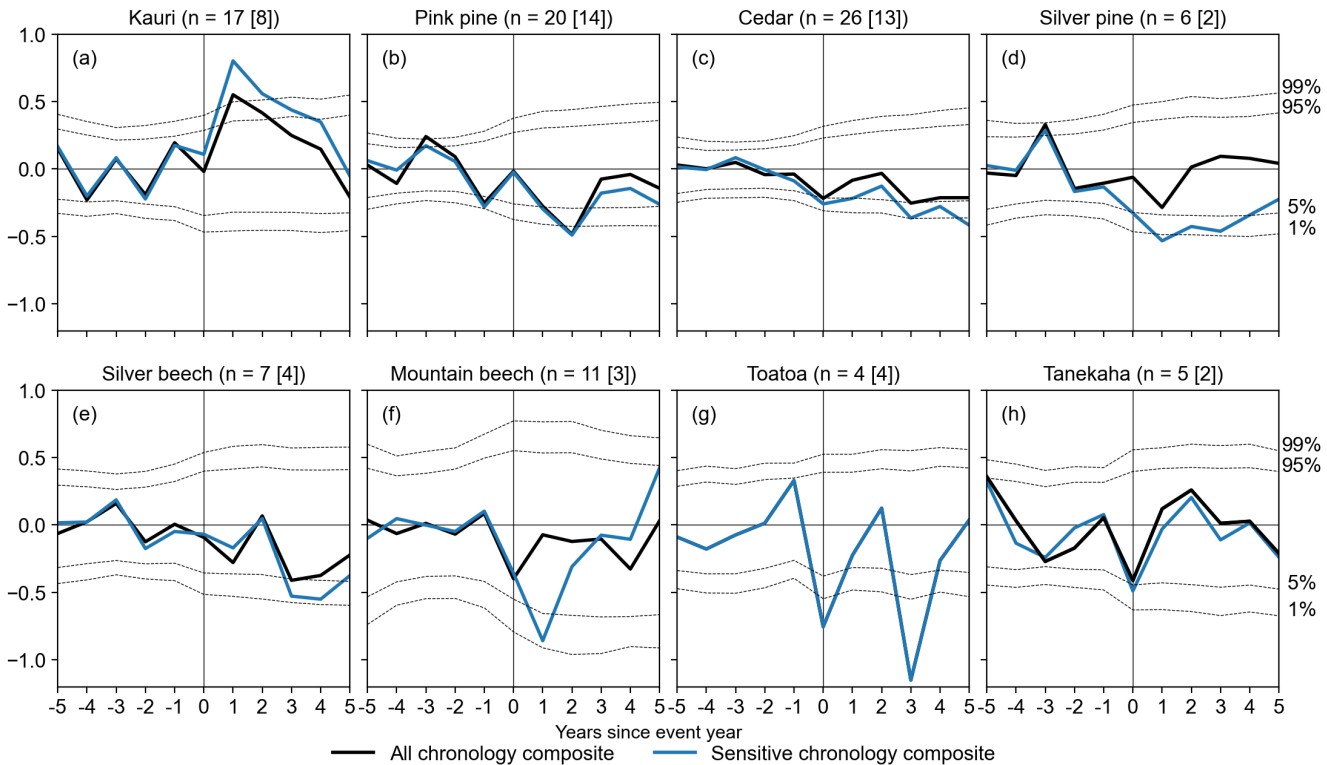

**Figure 3 – Mean chronology departures five years before and five years after 13 eruption years with SAOD > 0.08 (year 0), separated by tree species. The chronologies contributing to the species-wide composite are shown in black and the sensitive chronology composite in blue and the number of chronologies are shown in brackets/square brackets. Significance bands (dotted grey lines) are the 1st, 5th, 95th, and 99th percentile of 10,000 random samples of non-event years from the species-wide composite.**

## 3.2 Site based comparisons

The difference in species responses to volcanic events were investigated across six sites; three sites where cedar is co-located with pink pine and three sites where cedar is co-located with silver pine. Kernel density (violin) plots of the distribution of response to the 21-event series across all ring width series at each site are shown in Figure 4. No difference in response was observed between cedar and pink pine at Camp Creek or Mount French (Mann-Whitney U-test, p > 0.05), with both species displaying neutral responses to volcanic events (Fig. 4a-b). Both pink pine and cedar recorded significant negative volcanic responses at Takapari (Fig. 4c), with a larger response from cedar (Mann-Whitney U-test, p < 0.05). A significant difference in response between cedar and silver pine was identified at two of the three cedar-silver pine sites, Ahaura and Flagstaff Creek (Mann-Whitney U-test, p < 0.001; Fig. 4d-e). At both sites, cedar recorded a significant positive growth response compared to a neutral silver pine response. Neither species recorded significant responses at Mangawhero River.



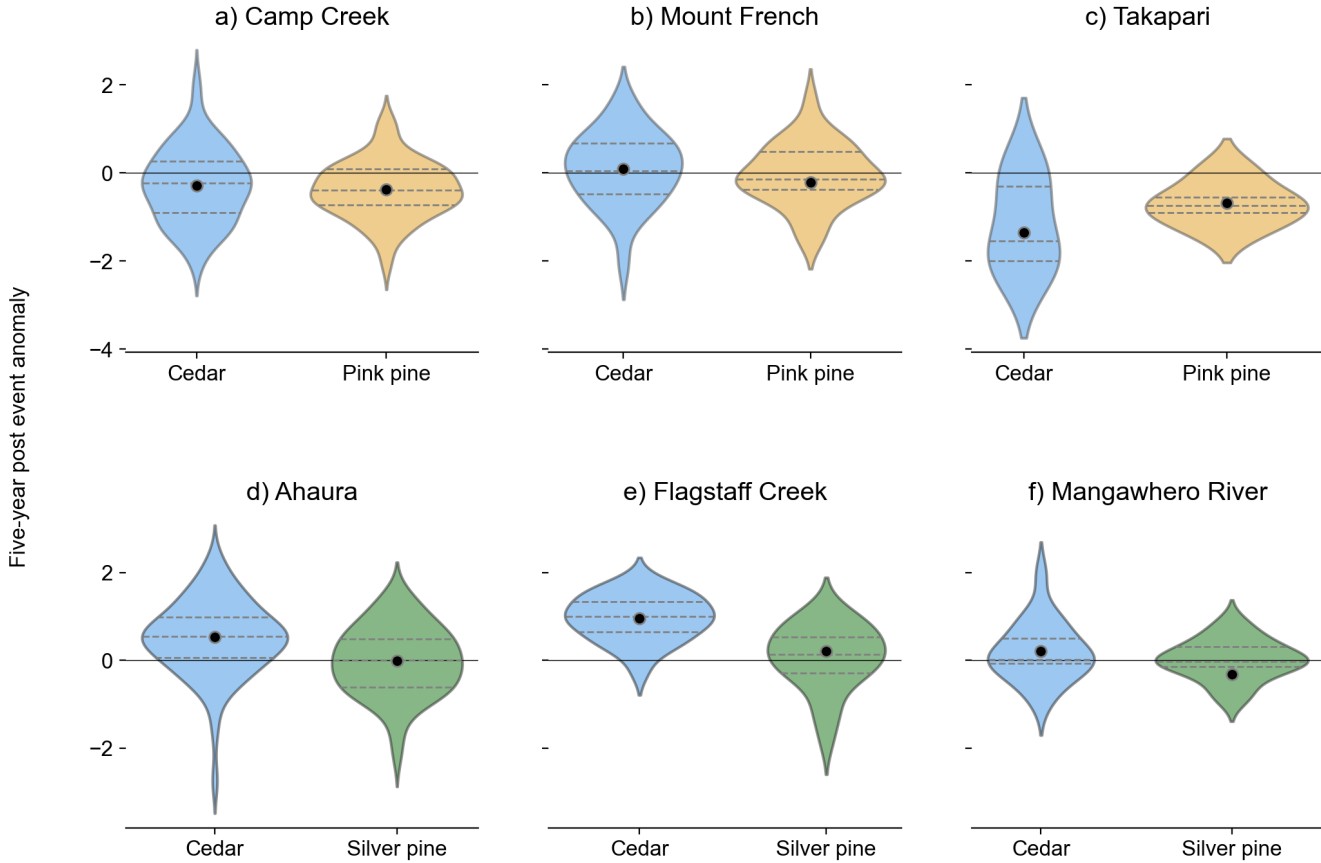

**Figure 4 – Kernel density (violin) plots of the five-year post event anomaly for the standardised ring width series contributing to the site chronologies, comparing cedar and pink pine (a-c) and cedar and silver pine (d-f). Dashed lines indicate the 25th, 50th, and 75th percentiles of the distributions, with the mean series response shown by the black dot.**

### 3.3 Variability in cedar response to volcanic events

Cedar ring-width series respond differently to volcanic events depending on their location (Fig. 4), with both very negative (Takapari) and very positive (Flagstaff Creek) responses recorded. We used K-means clustering via principal components analysis (Ding & He, 2004) to investigate whether this within-species variation could be explained by obvious factors like regional climate or elevation. Principal components analysis (PCA) was applied to the longest common time interval of the chronologies, 1732-1976 CE, and the first four principal components were retained.

Five chronology groups were identified via clustering (Fig. 5a), corresponding to differences in region and altitude. North Island chronologies were distributed in two groups. All chronologies are from montane to subalpine areas above 800 m, with groups, differentiated between coastal (G1) and inland (G2) locations. A single chronology from the north of the South Island was included in G2. West coast South Island chronologies were also distributed in two groups, differentiating between lowland (G3) and montane (G4) forest. The final grouping (G5) includes three chronologies from the dry eastern lowlands. Strong – but opposing - volcanic responses are identified via clustering. Significant lagged post-eruption growth reduction is identified





in groups G1 and G2 from montane to subalpine sites from the North Island, including Takapari, and the southern-most
grouping, G5. Group three (G3), which includes lowland chronologies from the northwest coast of the South Island, including
Flagstaff Creek and Ahaura, receives a growth benefit in the two years following an eruption similar to that observed in North
Island kauri. In cedar, we observed all three of the proposed temperate zone tree responses to volcanic events - positive,
negative, and neutral growth – all within a single species, highlighting the importance of site-based factors in determining tree
response in temperate zones.


**Figure 5 – a) Results of k-means clustering of cedar chronologies; b) – f) mean chronology departures five years before and after
eruption years (year 0), separated by cluster, and the 95th and 99th significance levels calculated by generating 10,000 random
samples of non-event years from the group composite. Map part a) made with Natural Earth free map data.**



### 3.4 Calibration and validation of the temperature reconstructions

Analysis of species-level temperature responses was limited to those chronologies extending to or beyond 1990; none of the chronologies from the two *Phyllocladus* species (toatoa and tanekaha) were therefore included in the analysis. The remaining six species show significant ($p < 0.05$) relationships with average New Zealand temperatures during individual months of the current growing season (Table 2, Fig. S2 – S6). Tree growth is only weakly correlated with average monthly temperatures, with $|r| < 0.3$ for most chronologies. Pink pine shows stronger correlations with summer temperatures, with r values of 0.4-0.6.

Pine pink is also significantly correlated to temperatures over the entire growing season, whereas the other species are seasonally restricted to peak summer months. Most species are positively correlated to current season temperatures, with wider ring widths associated with warm years; however, kauri and beech show an inverse relationship to temperature, with warm temperatures restricting growth. While there are significant correlations between individual chronologies and prior season temperatures, lagged correlations are weaker, and there is a less consistent within-species response than current season

correlations (Fig. S2 – S6).

**Table 2 - Summary of the correlations between autoregressively modelled tree-ring chronologies and New Zealand seven-station average temperature. Only correlations significant at the $p < 0.05$ level for multiple sites are reported. The + and – signs indicate the sign of the correlation.**

|  | Growing season (October - May) | |
| --- | --- | --- |
| **Species** | **Prior (year - 1)** | **Current** |
| Kauri | Nov (+) | Dec – Jan (-) |
| Pink pine | Dec – Jan (+) | Sep – May (+) |
| Silver pine | Dec (-), Feb – May (-) | Jan – Apr (+) |
| Cedar | Feb – Apr (-) | Sep (+), Feb – Mar (+) |
| Mountain beech | Feb (-) | Dec – Jan (-) |
| Silver beech | Feb – Apr (-) | Jan – May (+) |

The peak summer period was selected as the seasonal reconstruction target as the largest number of chronologies across species showed significant correlations with temperatures between December and February (Table 2). Selecting only those chronologies correlated at $p < 0.1$ with average DJF temperatures resulted in a predictor pool of 45 chronologies for reconstruction NZall, of which 25 showed significant volcanic impacts and were used to produce the reconstruction NZsens. Both New Zealand DJF average temperature reconstructions are shown in Figure 6 along with their instrumental fit over the

1911-1990 calibration period. There is good agreement between the reconstructions, with a Pearson r of 0.81 over the entire reconstruction period and 0.9 after 1750 CE. The initial, best-replicated nests, which cover the period 1790-1990, account for 67.5% and 58.2% of instrumental temperature variability for the NZall and NZsens reconstructions respectively. The minimum amount of variance explained over all nests is 44.1% for NZall and 34.3% for NZsens. The full calibration/validation statistics are provided in Supplementary Figures S7 and S8. For both reconstructions, RE values are positive over all nests from 1413-

1990, however CE values are only positive after 1520 CE in the NZsens reconstruction when calibrated to the early window (1911-1950). The declining instrumental data quality in the early period and the relatively few predictors retained for NZsens, are likely responsible for the negative values.



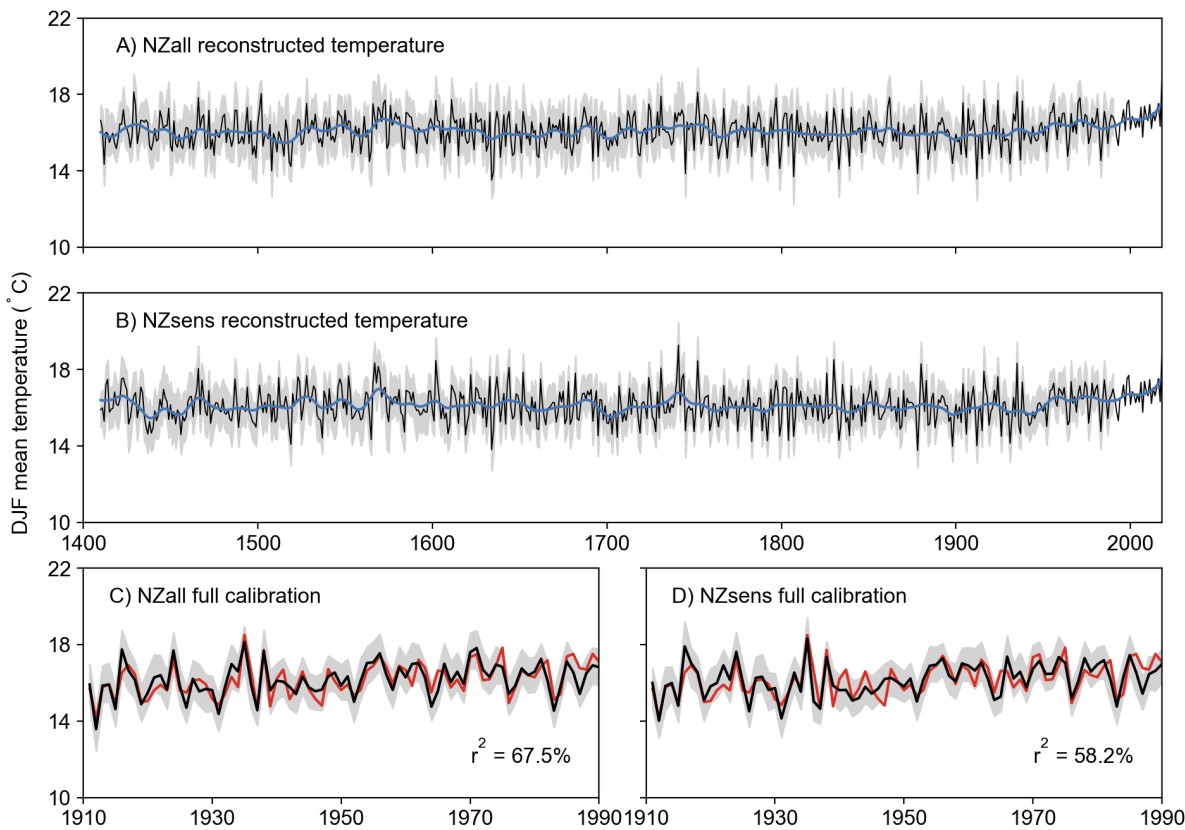

**Figure 6 – New Zealand average DJF temperature reconstructions. Unfiltered (black) and filtered (20-year spline; blue) mean DJF reconstruction with 90% uncertainty interval (grey) between 1400 and 2018 CE for A) NZall and B) NZsens. Reconstruction fit to instrumental temperature (red) over the full calibration period 1911-1990 for C) NZall and D) NZsens. The 90% uncertainty interval was calculated from 300 maximum entropy bootstrap replications.**

**3.5 Volcanic signals in the temperature reconstructions**

Figure 7 shows the results of the SEA analysis for the two New Zealand temperature reconstructions, for both sets of volcanic events, compared to the volcanic response of an ensemble of seven CMIP5 model outputs for the New Zealand region. For the 21 events with SAOD > 0.04, results are remarkably similar between the temperature reconstructions and the model ensemble. Both the timing and magnitude of the post-event anomaly, which is only significant in year t+1, are consistent across the models and reconstructions, as is the timing of the post-event recovery, which occurs in year t+2.

The response to the subset of events with SAOD > 0.08 shows larger year t+1 temperature anomalies for both models and reconstructions with the greatest difference in magnitude displayed by the modelling ensemble. Year t+1 anomalies are ~ 0.1 °C larger for NZall, ~ 0.2 °C larger for NZsens, and ~ 0.4 °C larger for the model ensemble than the 21-event anomalies. However, the mean model ensemble lies within the 90% uncertainty range of both reconstructions, indicating the difference in magnitude is not significant. The difference in post-event recovery is significant, with temperatures recovering by year t+2





in the reconstructions, but modelled temperature anomalies persisting in year t+2. This is the opposite result to many tree-ring based temperature reconstructions from the Northern Hemisphere which show lagged persistence compared to climate models due to biological effects in the ring-width series.

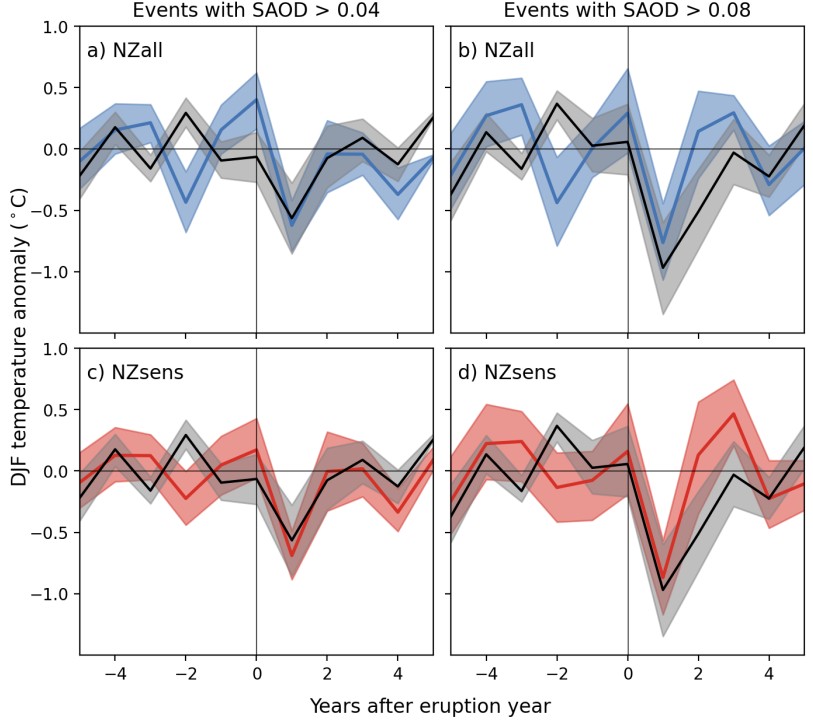

**Figure 7 – Mean anomalies five years before and after 21 eruption years with SAOD > 0.04 (left column) and 13 eruption years with SAOD > 0.08 (right column), for both the NZall (blue) and NZsens (red) reconstructions. The mean response from an ensemble of seven climate models to the same set of events is shown in black. The 90th percentile bootstrap confidence intervals were constructed from 1,000 replicates of either 15 or 9 event years at random.**

For both subsets of volcanic events the difference between the NZall and NZsens reconstruction response is minor. The

anomaly recorded by NZsens is 0.07 °C larger than NZall for the 21-event series and 0.1 °C larger for the 13-event series. The small difference between the reconstructions can be explained by the weightings applied to the chronologies in each reconstruction, with both reconstructions heavily weighted towards the same subset of chronologies. Four of the eight highest-weighted chronologies underpinning NZall are sensitive to volcanic events, and three of these are within the top four highest-weighted chronologies underpinning NZsens (Fig. 8). Thus, limiting NZsens to only sensitive chronologies had less impact on

post-eruption temperature anomalies than was expected.





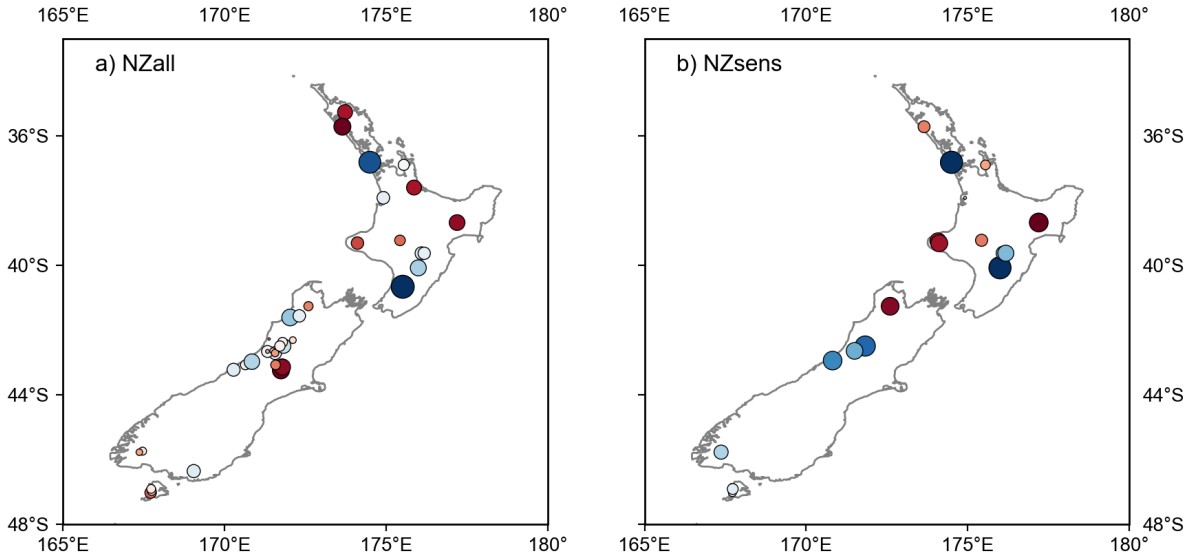

**Figure 8 – Distribution of tree rings used in a) the NZall temperature reconstruction and b) the NZsens temperature reconstruction. Colour and size represent the relative weighting given to each chronology in the multiple regression; red is negative and blue positive, and larger circles show greater weight. Made with Natural Earth free map data.**

**4 Discussion**

**4.1 Volcanic responses recorded by New Zealand trees**

Previous studies have not identified significant volcanic responses in Southern Hemisphere tree rings (Krakauer & Randerson, 2003; Palmer & Ogden, 1992) or in the temperature reconstructions based on them (Allen et al., 2018; Cook et al., 1992). We investigated whether Southern Hemisphere tree rings do record the impacts of volcanic eruptions using eight New Zealand

dendrochronological species. In contrast to previous studies, we found that volcanic events can be clearly identified in New Zealand ring widths, although some species are stronger recorders of volcanic signals than others. Unlike Northern Hemisphere high latitude and tree line sites, which tend to show a consistent reduction in growth due to volcanic dimming, no consistent response was identified across New Zealand conifer and *Nothofagaceae* species. Predominantly negative (pink pine, cedar, toatoa, silver beech), positive (kauri), and neutral (mountain beech, tanekaha, silver pine) responses were recorded.

As most New Zealand chronology sites have been sampled from localised areas of residual forest that are restricted compared to their natural distributional range, it is difficult to distinguish between species-related sensitivities to volcanic eruptions and regional climate factors that may control the response. In reality, it is the combination of biological characteristics including intrinsic species sensitivity, regional climate, and site-specific factors (e.g., soils, exposure to prevailing conditions) which determine the observed volcanic response. While necessarily simplified, we discuss below some possible explanatory factors

for the species-wide responses.





### 4.1.1 Stress tolerators and fast responders

The species-level results in Figure 3 and S1 clearly show two response types following volcanic events; a rapid but short-lived response, or a delayed response, which begins in year t+2 or t+3 but then persists over several years. The first response, demonstrated by mountain beech and kauri, we label here the "fast responder", and the second, shown by pink pine, silver
pine, cedar, and silver beech, the "stress tolerator" (after Grime (1979)). Many New Zealand conifers appear as typical stress-tolerators, which have adapted to growing in highly stressful conditions. As a group, they are longer-lived, slower-growing, taller, and markedly frost-tolerant compared to endemic angiosperms, and tolerant of poor soils (e.g., low nutrients and often poor drainage). Several species, including pink pine and silver pine, show an affinity for leached, low-nutrient, and water-logged soils (McGlone et al., 2017). The stress tolerators are characterised by narrow average ring widths and high biological
persistence (temporal autocorrelation) arising from carbohydrate storage or needle retention (see Table 1). Therefore, stress tolerators are slow to respond to changes in conditions such as volcanic dimming. The stress tolerator response resembles the response of high latitude Northern Hemisphere trees, although arctic trees display even greater lagged persistence, with suppressed growth for up to ten years following volcanic events (Krakauer & Randerson, 2003).

In contrast, the fast responders both respond and recover more quickly from a detrimental change in conditions (e.g., mountain
beech) or can rapidly capitalise on beneficial conditions (e.g., kauri). These species are relatively fast-growing, indicated by wider average ring widths than the stress tolerators, and have lower persistence (Table 1). Kauri could be considered a stress tolerator due to its affinity for poor soils, occurrence on ridges and slopes, and drought tolerance (McGlone et al., 2017); however, relative to other New Zealand conifers in this study, it is a fast responder.

The *Nothofagaceae* (beech) species-wide composites showed largely neutral responses to volcanic eruptions, suggesting most
chronology sites are not affected by volcanic dimming. Considering only those chronologies displaying negative responses, there is a clear difference between the species: silver beech acts as a stress tolerator, and mountain beech is a fast responder. Generally, New Zealand *Nothofagaceae* occupy sites with cooler mean annual temperatures than most conifers and are the dominant species growing at current tree lines. Silver beech is highly frost and exposure resistant, shade tolerant, and grows on extreme exposure sites (Manson, 1974; Stewart, 2002). In comparison, mountain beech is shade intolerant but has several
responses to abnormally cold temperatures, including rapid shoot growth and temporarily halting bud formation, which allow it to rebound quickly after a poor summer (J. Wardle, 1970).

In contrast to the subdued, persistent decrease in growth shown by the stress tolerator species, the initial decline in toatoa ring width in year t+0, then subsequent extreme decline in year t+3 and recovery by year t+4 closely resembles the boom-bust behaviour of the fast responders, but with several years' lag. The dominant climate response of toatoa is a negative correlation
to prior growing season temperatures and a strong positive correlation to summer temperatures two growing seasons prior. This results in a quasi-biennial pattern of wide and narrow rings, which has also been observed in other species of *Phyllocladacae* and may be related to a climate-triggered biennial flowering cycle (Allen, 1998; Ogden & Dunwiddie, 1982). Other potential explanations are foliage production followed by cladode senescence, or a mast seeding event, triggered by





multiple cool summers in New Zealand *Podocarpaceae* (Norton & Kelly, 1988). All three explanations suggest toatoa channels
resources to reproduction at the expense of cambial growth (Harper, 1977) following a climate trigger - perhaps reduced
summer water stress at low elevation North Island sites. More research on the ecology and life history of toatoa is needed to
confirm these possible mechanisms.

### 4.1.2 Site-based comparison of species

Sites with chronologies from more than one species provide the unique opportunity to directly compare species differences in
volcanic sensitivity whilst controlling for most other factors. The three species available for site-based comparison all showed
stress tolerator responses to volcanic eruptions. Pink pine and cedar often grow together in mixed stands. Both species are
sensitive to temperature, although pink pine has a maximum correlation to late summer temperature, whereas cedar responds
to conditions in the winter prior to the growing season and in spring (Fenwick, 2003). Whilst neither species at the two sites
on the west coast of the South Island showed a significant volcanic response, both species showed a decline in ring width post-
event at Takapari, in the North Island ranges. There is little difference in altitude between the three sites, but substantial
differences in regional climate. The South Island west coast (Westland) of the Southern Alps is the wettest part of New
Zealand, with annual rainfall >2,000 mm and low precipitation variability throughout the year. Westland trees experience little
water stress, and chronologies from Westland sites are not sensitive to precipitation (Fenwick, 2003). In comparison, annual
rainfall is around 1,000 mm at Palmerston North, close to the Takapari site, and chronologies respond positively to late summer
precipitation (Xiong et al., 1998). Chronologies from the Takapari site may be responding more strongly to decreased summer
precipitation due to a change in circulation patterns following a volcanic event (McGraw et al., 2016; Salinger, 1998), or to a
local variation in the cooling response. The difference in responses between pink pine and cedar was also significant at
Takapari, with cedar showing a stronger response. However, only 12 pink pine series were available for Takapari, far fewer
than any of the other sites or species. Comparison over additional sites is required to determine whether there is a difference
in the sensitivity of pink pine and cedar to volcanic dimming.
Differences between the cedar and silver pine responses can be observed at two of the three sites, with cedar showing greater
sensitivity to volcanic eruptions. At both sites, the cedar mean response is positive, whereas the silver pine response is very
close to 0. Silver pine is primarily found in the moist, temperate, low-elevation forests of the west coast of the South Island.
It is a shade-tolerant species that grows in highly competitive closed-canopy forests on infertile, poorly-drained or water-
logged soils (Cook et al., 2002; P. Wardle, 1977). It is exceptionally slow-growing and shows little year-to-year variability in
ring width (Table 1). Thus, it is unsurprising that volcanic effects were more readily identified in cedar at the Ahaura and
Flagstaff Creek sites. Figure 2 shows a neutral species-level volcanic response from silver pine, with only the two southern-
most sites recording significant responses. At these sites we lack cedar chronologies for comparison.





### 4.1.3 A Kauri growth benefit

An interesting result of this study is the strong positive species-wide response of North Island kauri to volcanic events. Over 70% of the kauri chronologies recorded a small but significant increase in ring width in the year following a large eruption (SAOD > 0.04), indicating a growth benefit from volcanism. Previous studies of kauri climate-response function have shown that growth is not primarily related to temperature but is enhanced during cool, dry years, with the strongest (negative) correlation to austral spring temperatures (Buckley et al., 2000; Ogden & Ahmed, 1989). Ring growth is thus enhanced during

El Niño events, which result in cool, dry spring conditions in northern New Zealand, and kauri has been successfully used as a proxy for the El Niño-Southern Oscillation (Fowler et al., 2008).

The mechanism behind this counter-intuitive relationship remains largely unclear, although it has been proposed that reduced cloud cover during El Niño events may benefit kauri growth via increased insolation (Fowler et al., 2000). The opposite conditions follow a volcanic eruption, with reduced direct insolation and increased diffuse insolation due to volcanic aerosols.

The commonality between the two sets of events is cooler-than-average spring/summer temperatures. This suggests that kauri may capitalise on a decrease in summer evapotranspiration during both El Niño events and following significant eruptions. Maximum kauri growth occurs during spring, with large declines in growth rate over the peak summer months (Fowler et al., 2005). Reduced spring/summer moisture stress may delay the cessation of growth, resulting in wider annual rings (Palmer & Ogden, 1983).

Many observational and modelling studies propose a link between large tropical volcanic eruptions and sea surface temperature variability in the tropical Pacific, with El Niño-like conditions more likely in the year following a significant event (Adams et al., 2003; Christiansen, 2008; Emile-Geay et al., 2008; Khodri et al., 2017; McGregor et al., 2010; Miao et al., 2018), although this link is not always identifiable in the paleoclimate data (Dee et al., 2020). The three eruptions included in this analysis since 1900 co-occurred with an El Niño event, and the 1982/83 El Niño is one of the largest on record (Santoso et al., 2017).

While we do not wish to debate the eruption-ENSO response as part of this study, these potential interactions complicate our analysis of the volcanic signal in kauri.

In an attempt to distinguish between the effects of El Niño events and volcanic eruptions on kauri growth, we repeated the SEA analysis removing the three volcanic eruptions since 1900. A smaller composite ring width anomaly was recorded without the three events, but the response remained significant in year t+1 (Fig. S9). To test the potential follow-through impact of the

kauri response to El Niño events on the temperature reconstructions, we removed the ENSO component via linear regression of the Southern Oscillation Index on the reconstructed temperature series. There is a negligible difference between the original and ENSO-adjusted temperature reconstructions for all volcanic events between 1880 and 1990 CE, except for the response to El Chicon in 1982, which is much larger in the unadjusted reconstruction (Fig. S10). Based on currently available data, we cannot confidently discount that the kauri growth benefit identified in year t+1 may be a secondary response to changes in

tropical Pacific sea surface temperatures following a large eruption. However, this is unlikely to have a large impact on the post-event anomalies identified in the temperature reconstructions.



### 4.2 Site-related volcanic responses

Based on the results in Figure 4, it is evident that site-related factors can have more control over volcanic response than the difference between species. This finding was further explored using k-means clustering of the widely distributed cedar

chronologies (Fig. 5). Altitude and latitude are important explanatory factors for tree growth, as together they represent the relative importance of temperature and water stress at a site. Trees at their altitudinal or latitudinal limit are more likely to show sensitivity to temperature, including volcanic cooling. Temperature-limited high-altitude cedar sites at or near the timberline in the North Island ranges show a significant decline in tree growth following eruptions, as did coastal sites at higher latitudes (~ 46 ºS).

Low-elevation trees are more likely to experience summer water stress than their high-elevation counterparts, and thus sites near the low altitudinal/latitudinal limit of a species may also contain important temperature information. We observed a significant increase in cedar growth at low-elevation sites on the northern South Island (Group 3) in response to volcanic cooling, which is assumed to reduce evaporative demand over the summer. These chronologies also display a negative correlation (not significant at $p < 0.05$) to average New Zealand summer temperatures in contrast to the significant positive

correlation of high elevation sites (Figure S6) supporting the finding that water stress (i.e., soil moisture) is a limiting factor at these sites. Thus, we find that both high and low-elevation cedar stands can reliably record volcanic signals, provided the site experiences sufficient temperature or moisture stress.

Tree growth of species at different sites is limited by a variety of environmental factors, of which temperature and soil moisture are only two (Fritts, 1976). For many New Zealand species, little is known about what types of sites might accentuate these

factors and thereby enhance the climatic sensitivity in the tree-ring series (Dunwiddie, 1979). Although the overall Group 2 cedar response was significant, not all high-altitude sites recorded a volcanic signal. Considering the location, aspect, forest characteristics, and soil type at individual sites, we find that exposure to prevailing conditions is the key explanatory variable for the within-species response for those sites near the altitudinal range. Sites that record a significant growth response have high exposure to prevailing winds and are more sensitive to abnormally low growing season temperatures. In contrast,

chronologies from sites characterised by undulating ridgelines and more continuous forest showed a neutral growth response. North Island kauri also demonstrates how exposure affects chronology response to eruptions. For kauri, sites with a strong positive response to volcanic eruptions are seaward sites exposed to prevailing wind conditions or sites limited by poor underlying sediment substrates. In comparison, sites that showed little volcanic response were those on the leeward side of the coastal range, which are buffered by inland microclimate effects. These sites experience mesic conditions and less water stress

during the summer; therefore, we expect they receive less benefit from volcanic dimming reduced evaporative demand, resulting in a neutral response.



### 4.3 Implications for temperature reconstructions

This study shows that large volcanic eruptions impact New Zealand climate and that tree rings can reliably record this impact.
However, the volcanic response recorded in New Zealand dendrochronological species is complex and negative, neutral, and
positive responses are possible depending largely on site-related factors. We produced two new multispecies reconstructions
of New Zealand average summer temperatures to test whether chronology selection would impact whether a volcanic response
could be identified in the reconstructions. All chronologies significantly correlated with DJF temperature were included in
reconstruction NZall, but reconstruction NZsens was limited to those chronologies with an individual significant volcanic
response, under the hypothesis that NZsens would therefore show a larger magnitude of post-event cooling. We observed
significant temperature anomalies in the year following an eruption in both our reconstructions, regardless of the volcanic
threshold selected. The post-eruption temperature anomalies in both reconstructions are comparable to the limited observations
of Salinger (1998) for instrumental temperatures.

Unexpectedly, we found almost no difference in the volcanic response between the reconstructions. While in each case NZsens
showed a slightly greater magnitude of cooling than NZall, the confidence intervals around the ensemble responses for both
reconstructions overlap, indicating that the difference is not significant. As shown in Figure 8, both reconstructions are heavily
weighted towards the same subset of chronologies; thus, limiting NZsens to only sensitive chronologies had less impact on
post-eruption temperature than expected. In developing NZsens, we used a "volcanic sensitivity" threshold based on the SEA
result's significance ($p < 0.05$). In doing so, we reduced the size of the predictor pool, which reduced the strength of the
reconstruction, particularly over the initial 100 years when there were relatively few predictors (Fig. S8). The loss of
reconstruction strength outweighs the small increase in volcanic sensitivity in NZsens and we conclude that it is not beneficial
to restrict the predictor pool in this instance. Another factor leading to the minimal difference between the reconstructions is
that many volcanically sensitive chronologies, particularly kauri, were cored before 1990 and therefore not included in either
temperature reconstruction. These sites should be updated with priority for future studies of volcanic impact in the Southern
Hemisphere.
Recent work investigating the reasons for differences between climate model and proxy reconstructions of post-event
temperature anomalies in the Northern Hemisphere (Zhu et al., 2020) found that these differences can be minimised by
focussing on the growing season rather than annual temperatures, undertaking regional rather than hemispheric analysis, and
resolving biological persistence. The results of this study support these conclusions. We compared modelled and reconstructed
temperature anomalies over the New Zealand region for DJF – peak growing season in the Southern Hemisphere. We found
no difference between the magnitude of the year t+1 anomaly for either the 13- or 21-event composites, with reconstruction –
model anomalies < 0.12 ºC for both sets of events. A criticism of temperature reconstructions based solely on ring widths is
that biological persistence in treeline conifers decreases the abruptness and magnitude of volcanic cooling. In the Northern
Hemisphere, more emphasis is now being placed on maximum latewood density (MXD) or mixed MDX and ring widths for
investigations of volcanic cooling (Wilson et al., 2016; Zhu et al., 2020). For this study, we focussed only on ring widths as





few investigations of alternative wood properties have been undertaken in New Zealand (Blake et al., 2020; Xiong et al., 1998). Unlike Northern Hemisphere studies, our ring width temperature reconstructions show no increased persistence in temperature anomalies following eruptive events compared to the climate model ensemble. Ring widths from New Zealand conifers therefore appear suitable for volcanic investigations.

## 5    Conclusions

Very few studies have considered whether volcanic signals are identifiable in tree-ring chronologies from the Southern Hemisphere. We investigated whether volcanic events could be identified in New Zealand tree rings, using data from eight dendrochronological species. In doing so, we set out to answer three questions: 1) can volcanic signals be identified, 2) are there species-level differences in volcanic signal strength, and 3) does proxy selection impact the magnitude of post-volcanic cooling temperature reconstructions from tree rings.

In answering the first two questions, we found that New Zealand dendrochronological species are good recorders of volcanic dimming, but that response varies across species. The magnitude and persistence of the species-wide volcanic response can be broadly linked to plant life history traits. The larger magnitude and more immediate responses are recorded by the "fast responder" species, such as mountain beech and kauri, and more delayed but persistent responses are recorded by the "stress tolerator" species such as silver pine. In general, volcanic events can be more readily observed in the ring widths of fast

responder species, which should be prioritised for future regional or hemispheric studies. Unfortunately, the paucity of information on the ecology of many New Zealand species limits our understanding of how species allocate resources to processes other than cambial growth in response to short-term changes in climatic conditions.

The volcanic response of New Zealand trees is complex, with positive, negative, and neutral responses identified sometimes within the same species group. For sub-alpine sites, this finding is not dissimilar to previous studies of temperate zone Northern

Hemisphere species. We found that site-related factors have greater control over displayed volcanic responses than species. The altitude of the site with respect to the species altitudinal limit, and exposure of the site to prevailing conditions, were the most important factors determining whether a chronology volcanic response could be identified. In some cases, sites near the lower altitudinal limit of the species were also strong responders, provided summer moisture stress was a limiting factor. This indicates that studies intending to utilise tree rings to investigate regional volcanic cooling should carefully consider the

characteristics of the sample site. While valid for all dendrochronological studies, it is particularly important for identifying volcanic signals as we find that the range of temperature-sensitive sites is greater than the range of volcanically sensitive sites. In answer to the final question, we developed two new reconstructions of New Zealand summer temperature to investigate whether proxy selection impacted the magnitude of post-volcanic cooling. There was little difference in the post-event anomalies, suggesting limiting the predictor pool for volcanic sensitivity is not necessary when targeting average growing

season temperatures. Both reconstructions showed temperature anomalies remarkably consistent with studies based on instrumental temperature, and with the ensemble mean response of CMIP5 climate models. Based on the results here, New Zealand ring widths are reliable indicators of regional volcanic climate response.



More broadly, the findings of this study have important implications for the development of future tree-ring temperature reconstructions from the Southern Hemisphere. Dendrochronologists should be cognisant that chronology selection may impact the magnitude of volcanic cooling identified in temperature reconstructions. This is particularly important for subregional reconstructions or reconstructions based on chronologies from a single species. The results are also important for tree-ring or multiproxy hemispheric temperature reconstructions, which often incorporate species-specific "master" chronologies (i.e., composite chronologies developed from across many sites) into their predictor pool. As shown in this study, the compositing process can result in reduced volcanic signals when more than one type of response (i.e., positive, negative, or neutral) is recorded by a single species. However, as most New Zealand species-level composites show significant volcanic responses, temperature reconstructions based on composite chronologies should also show the influence of volcanic eruptions.

## Competing interests

The authors declare that they have no competing financial interests.

## Author contribution

PAH and JGP conceptualised this study, JGP curated the data and PAH undertook the analysis. PAH wrote the manuscript with contributions from all authors.

## Financial support

PAH is supported by an Australian Government Research Training Scholarship and the UNSW Scientia PhD Scholarship Scheme. FJ is supported by the UNSW Scientia Program. Further support was provided by the ARC Centre of Excellence in Australian Biodiversity and Heritage (CE170100015).

## Data availability statement

All data and software used in this study are publicly available. The New Zealand 'seven station' temperature series was downloaded from NIWA at https://niwa.co.nz/. CMIP5 climate model outputs were downloaded from https://esgf-node.llnl.gov/projects/cmip5/. The Southern Oscillation Index time series was sourced from https://www.ncei.noaa.gov/. Meta data for the tree ring chronologies can be found in the supplement to Palmer et al., (2015), and the raw ring width series can be downloaded from the International Tree Ring Data Bank at https://www.ncei.noaa.gov/. Superposed epoch analysis was undertaken using R code published on Mendeley Datasets with DOI 10.17632/8p7y29hz5h.1.

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
