# Peer review of "Do Southern Hemisphere tree rings record past volcanic events?"

_Climate of the Past, 2021_

## Author Comment (AC1)

**General comments**

This paper led by Higgins examines the ability of eight New Zealand tree species to reflect volcanic dimming. As the authors point out, little is known of the impacts of past volcanic eruptions on Southern Hemisphere climate, and there is a discrepancy between models and palaeo-data. The authors found that there are differences across and within species in terms of their apparent reaction to dimming. The authors also present a summer temperature reconstruction from the tree-rings that reflects influence of dimming. This same response is detected in the 7-model ensemble response. Previous studies on Southern Hemisphere trees have not identified a volcanic impact. This study is therefore of considerable interest and importance. Overall, the manuscript is quite well written, although some work is required to improve clarity and succinctness, and to better emphasise main findings.

Thank you for your thoughtful and detailed review. In our opinion, your review raises two major points to be addressed and several more minor points. The two major points, which are reiterated over several questions, are:

1. The use of regional dimming index for event selection and whether this then biases the results of our study, explaining why previous studies have not found a Southern Hemisphere volcanic temperature impact.

2. Why biological persistence in ring-width records does not appear to impact the volcanic signal in the temperature reconstruction, particularly compared to Northern Hemisphere studies.

All answers are provided below the questions in blue text, or, where appropriate, reference the response to another review that has asked the same question. Please note we have renumbered your points because the original numbering was corrupted in the uploaded version.

I wonder if a rearrangement of the material would help emphasise the key findings of this study a little better, better follow on from the introduction and also provide a basis for stronger and more substantial conclusions. One suggestion might be to start with showing a response in a new [more fully described] temperature reconstruction (or reconstructions) composed of a multi-species network of temperature sensitive chronologies to regional volcanic dimming, and a comparison with the CMIP ensemble before delving into the details of species used and species-level responses? The reason I suggest this, is that the introduction (l. 40 – 49, 65) seems to indicate that the model-paleo discrepancy will be an important aspect of the paper, but this doesn't really come through (might the choice of a regional dimming index be relevant? – see below). I think that finding a volcanic response is the first 'big' result of the study. I am guessing the authors view it as more useful to build the case from the sites/species first and then to look at the temperature reconstruction but nevertheless ask that they carefully consider what aspects of their work deserve greatest emphasis.

We believe there are two 'big' results from this study, and both deserve to be highlighted in the paper.
a) opposing volcanic signals can be identified in chronologies from the same species, and b) there is a clear Southern Hemisphere volcanic dimming signal.

While the first result may be predominantly of interest to dendrochronologists and the second to the wider paleoclimate community, we don't believe either is of greater importance. As you have acknowledged, the paper is structured to build a case for temperature reconstructions from the site/species information. It is also structured this way to highlight both significant results. We hope that in resolving some of the other issues you have raised and revising the discussion in response to your and the other reviewers' questions, the conclusions may be emphasised without a paper restructure.

The authors' use of a regional dimming index relevant to New Zealand latitudes/longitudes rather than a selection of events based on eruption magnitude may be a key reason for their results. I think this needs to be discussed in more detail – it seems to be the elephant in the room when the authors are focussed on reasons for differences in responses amongst species. The fact that a relationship with volcanic eruptions has been identified in temperature-sensitive Southern Hemisphere trees is highly newsworthy. Why/how did the authors find this when other studies haven't? It would be good to place considerably more emphasis on this in the Discussion/Conclusions.

To test whether the event selection is the reason why our study finds a significant volcanic response when others haven't, we looked to the studies of Allen et al. (2018) and Cook et al. (1992, 2002) both of which produced a temperature reconstruction, assessed them for volcanic impact, and concluded the relationship between reconstructed temperature and the volcanic response was not clear/significant. The

November-April temperature reconstruction of Cook et al. (2000), based on a single Huon pine chronology from Mt Read, Tasmania, was downloaded from the NOAA paleoclimate database. We then ran SEA using both the 21-event list and 13-event list and plotted the results. The figure below shows a significant response in year t+1 to the largest volcanic subset (SAOD > 0.08) but no response to the entire event list (SAOD > 0.04). Therefore, while a volcanic response is evident in the Mt Read reconstruction, it is more sensitive to the magnitude of the volcanic dimming than the New Zealand reconstruction. Without undertaking a detailed analysis, there are several reasons why this may be the case – Tasmania may be more influenced by the maritime climate than New Zealand, Huon pine may not be a strong responder to volcanic events, using multiple chronologies in our reconstruction may strengthen the volcanic signal compared to the single-chronology Mt Read reconstruction, and/or the difference in target seasons (summer versus growing season) may be an important factor.

[Figure]

**Figure 1 - Mean anomalies five years before and after 21 eruption years with SAOD > 0.04 (left column) and 13 eruption years with SAOD > 0.08 (right column), for the Tasmanian temperature reconstruction of Cook et al. (2000).**

Why no response was identified by Cook et al. (1992) is impossible to analyse as the paper does not provide an event list or specify the methodology used to assess the link between volcanoes and reconstructed temperature. However, it can be seen in Figure 1 that only events resulting in quite significant Southern Hemisphere volcanic dimming can be identified in the Mt Read reconstruction.

Event selection is therefore likely to have contributed. In the last 20 years, many research efforts have focused on reducing the uncertainty in the dating of volcanic events (e.g., Toohey & Sigl (2017)).

Therefore, the influence of volcanic dating errors in the analysis of Cook et al. may also be a factor in their lack of a significant response.

Allen et al. (2018) produced two reconstructions of Tasmanian summer temperatures, one based solely on *Lagarostrobos franklinii* and the other based on several endemic conifer species. It would be ideal for testing these reconstructions against our event list, as they targeted the same season (DJF) and used the same reconstruction methodology as our current study. These reconstructions are not publicly available but can be requested from the authors and are an obvious aspect of a follow-up study.

See also the response to Question 30 regarding previous NZ temperature reconstructions.

The authors rightly mention the moderate temperature response of the New Zealand species. This also applies to the ring widths of other Southern Hemisphere species. Do the authors consider that their use of the regional dimming index might 'compensate' somewhat for this moderate temperature response?

Using a regional dimming index to select events rather than an eruption magnitude almost certainly has contributed to identifying volcanic signals in this study. This was the intention of the event selection 90    method, and we believe it was the right choice for this study. However, we don't think of the selection method as 'compensating' for a moderate temperature response. All tree ring-volcano studies select a threshold for which events to include, and this is known to be a significant source of uncertainty. Whichever the chosen threshold, studies aim to select events that would have resulted in a climate response in their study region/hemisphere. We used two different event thresholds and bootstrapped 95    confidence intervals to account for this uncertainty in our results. The 90th percentile bootstrapped confidence intervals show that event selection could substantially impact the observed temperature response.

Please also see the response to RC2, where we demonstrate the effect of using a hemispheric rather than a regional dimming threshold.

In relation to the more detailed analysis of species and sites, it would also be useful to show more information on the actual sites. It is not really sufficient to state that the meta-data for sites can be found in Palmer et al. 2015. It is difficult to adequately consider some of the points made by the authors in the discussion without having the sites put into context much earlier. For example, the information in Table 2 (along with references to the supplementary Figure 2) could be presented in Section 2.1. A 105    summary (possibly pictorial and perhaps in the Supplementary?) of the various species' sites by altitude/location would also be very helpful to better guide the reader through the results/discussion. Would such a figure help when providing some detail on which sites within a species did not have a strong volcanic response (Section 3.1)? Are there common factors – like altitude for example – that play a role in nonsignificant response within species? Could it be linked in any way to the reason for 110    previous studies not finding a relationship with volcanic eruptions? (e.g. the authors discuss elevation and latitude).

We are happy to include the site metadata to a table in Supplementary – this was also requested by RC1, and we have included the proposed example in the response to RC1.

A summary of the temperature sensitivity of the different species can also be added to Section 2.1 with 115    reference to the supplementary figures. However, there are limits to the details available or that can be provided (e.g., exhaustive site details were not always recorded).

Also, the authors comment in the conclusions that only a subset of the temperature sensitive chronologies show a response to volcanic eruptions. Figures S2-6 show that a number of the chronologies that are not temperature sensitive.

a) If the argument is that volcanic eruptions affect temperature and it is this that then impacts radial growth, why not use this information to exclude chronologies from the reconstruction and/or the analysis for a volcanic signal in the first place?

While there are broadly consistent temperature sensitivities for each species group, as the reviewer highlights, different chronologies show different strengths in these relationships (Figures S2-S6 –
please note only chronologies extending until 1990, and thus considered for the temperature reconstructions, were plotted on these figures).

Considering the full suite of 96 chronologies in this study, only four (1HUI.r, 1MOE.r, 1MWL.r, 8WKT.r) shows no significant ($p < 0.05$) correlations to NZ average temperature in any month over the two growing seasons. As this is the likely criterion to be used to exclude chronologies from
volcanic analysis, the change to the results would be negligible.

    b) Some better links between this information and discussion of which sites do and do not show a volcanic response (or show a range of responses – i.e. cedar) may be warranted. Ditto in terms of positive/negative responses – does a dominant current season [positive] temperature response equate to a negative response to volcanic eruptions? Does a dominant prior season
[negative] temperature response equate to a positive response to volcanic eruptions? (for eg)?

A general consideration of these points can be added to the discussion, although with the small volume of data available, it is not possible to tease out all these relationships in full. Figure 2 summarises the temperature-volcanic response relationships for the three species with the largest numbers of chronologies: kauri, pink pine, and cedar. The species responses are clearly identified,
positive volcanic response for kauri, negative response for pink pine, and mixed response for cedar. Note that the significant positive volcanic response in pink pine is TOS.r, a site with substantial disturbance, so confidence in this result is lower.

Considered over both growing seasons, there appears to be a relationship between the strength of the temperature response and the strength of the volcanic response at most sites (as expected). The
correlation between current growing season temperature and volcanic response is complex, with significant relationships in all four quadrants. The relationship between previous growing season temperature appears more straightforward, with a negative temperature response associated with a positive volcanic response and vice versa.

[Figure]

**Figure 2 – Summary of the relationship between sensitivity to temperature and volcanic response for three species: kauri, pink pine, and cedar. Left: maximum temperature correlation in any month of the prior growing season against maximum volcanic response in the five years following an eruption for the 13 largest events. Right: same plot but for the current growing**

**season. Filled markers indicate that a site has a significant temperature correlation (p < 0.05) and a significant volcanic response (< 5th or > 95th percentile of bootstrapped responses). Open markers are not significant for either the temperature or volcanic response, or both.**

c)   Any comment on seasonal window of temperature response and its relevance (or not) to volcanic response?

The seasonal sensitivity of tree growth is potentially an important determinant of the volcanic response, but we don't have sufficient evidence from the results of this study to support a discussion. It would be very interesting to investigate whether the whole-of-season sensitivity of pink pine causes more or less sensitivity to climate disturbance compared to the other species with narrower temperature sensitivity, but this is a question for future research. A greater number of (updated) sites with two or more species (i.e., pink pine and cedar) would allow a comparison of the role of seasonal sensitivity largely without other confounding factors.

Loosely linked to this point, the authors make the case that lower elevation sites have a temperature response related to moisture stress (l. 432-434, 503). While this is not an unreasonable suggestion, the authors need to be more careful about how they state this – they have not shown data to support the statement, so comments should be more cautious when it is mentioned.

Yes, the temperature-moisture stress response at low elevation sites is a hypothesis and should be qualified as such. Please also refer to the response to RC2, where we provide additional evidence for the kauri-moisture stress hypothesis based on dendrometer band studies.

At face value, there seems to be an inherent contradiction in the authors' discussion around 'stress tolerators' and delayed responses and their later comments about the lack of a lagged response to volcanic eruptions in the temperature reconstruction. It is important to clarify this given that the memory in tree-rings has been found to be an issue in the response of Northern Hemisphere trees to volcanic eruptions. The authors should carefully consider what they are implying in their discussion of lagged response (as shown in Figure 3 and S1) as opposed to their comments about the 'lack of memory' in the temperature reconstruction. Why are there apparently lagged responses of varying magnitude across the chronologies that are not apparently reflected in the reconstruction? Some careful consideration needs to go into this.

Firstly, the suite of tree rings used in this study have lower biological persistence compared to published Northern Hemisphere ring-width records (Esper et al., 2015). 80% of the predictors used to develop the NZall reconstruction only have significant lag-1, or lag-1 and lag-2 autocorrelation after standardisation. So, while we see lagged effects after eruptions in the composite for the stress tolerators due to persistence, there is less lag compared to arctic trees. Please see also the response to RC1.

Reconstruction methodology also plays an important role in the persistence of the final temperature reconstructions compared to the predictors used. We refer to the study of Büntgen et al. (2021), which compares 15 temperature reconstructions developed by different research groups with different methodologies but using the same set (or a subset) of chronologies. AR1 persistence in the final reconstructions varied from < 0.4 to > 0.9 due to methodological decisions, and there was a substantial variation in response to volcanic cooling between the reconstructions (Figure S2, their study). The important elements of our reconstruction methodology which likely contribute to the final autocorrelation structure are a) pre-whitening of both the tree rings and temperature data prior to reconstruction and b) testing predictors for significance in both year t and t+1.

Why were the specific 7 CMIP models selected (Table S2)? Why not other models?

These were the models that could be freely downloaded from the ESGF@DOE/LLNL, and met the condition of having both a past1000 and historical run.

There is quite a bit of repetition (almost the same sentences in some cases) between the Results and Discussion. This should be minimised, especially given that the authors are presenting a range of interesting results across and within species groups, and a temperature reconstruction and its response that is compared with a model ensemble. It is important to draw the threads together as coherently as possible.

Agreed.

A minor point: the title implies Southern hemisphere, but the study is focused on NZ. Perhaps it would be pertinent to include "A case study from New Zealand" or similar in the title.

Agreed.

*Specific comments*

Introduction

1. 47-9 Might seasonality of signal matter?

   Yes, see the response at line 165.

2. 52-5 "Tree growth…." Yes, but this almost sounds like the vast majority of the SH trees should be ruled out simply based on location and lack of serious elevation. It also seems to differ from what some of the results suggest (low elevation sites in mid-latitudes apparently sensitive).

   This is not the intention of the paragraph. We wish to highlight that it may be harder to identify volcanic signals from less temperature–limited trees, as indicated by similar studies in the

Northern Hemisphere.

3. 66 – 7 – This sentence doesn't follow previous. Why would understanding whether a site is likely to have a volcanic response necessarily be relevant for studies of hydroclimate?

*We are not sure to which sentence the reviewer is referring. Sentence 66-67 reads, "This knowledge will benefit future studies of hemispheric temperatures and help identify which species and/or regions should be prioritised for future proxy development."*

4. 73-74. Reword a little – awkward to read.

*A proposed change to:*

*Land clearing has resulted in the loss of most lowland forests and nearly all forests from the eastern drylands of New Zealand. The most common remaining forest types are wet conifer-broadleaved forests and montane to alpine southern beeches (Nothofagaceae) dominant forests (McGlone et al., 2017).*

Methods

5. 104 maybe reword this first sentence slightly to improve clarity.

*A proposed change to:*

*Event selection is a significant source of uncertainty in tree-ring studies of volcanic cooling.*

6. 113 10 and 18 eruptions. On next page on l. 126 and also l. 118, 13 and 21 events? Seems to be an error here? In any case, this is confusing.

*This can be clarified. 10 and 18 refer to the number of events between 1400 and 1900 CE, including the three events subsequently happening in the 1900s, bringing the event lists to 13 and 21.*

7. l.116 "Between 1900 and 1990, we selected the three largest.." Where these the largest based on the same criteria for selecting the historical eruptions? (ie. based on the regional dimming index?)

*The dataset we used for the latitudinally modelled SAOD (Toohey and Sigl, 2017) doesn't extend past 1900. Therefore, event selection from 1900 was not based on the same dimming index. Please refer to the response to RC2 to see the impact on the results from using a different dataset to set the thresholds.*

8. 128 "Species-level…". This again makes me wonder if it would be wise to first screen out those sites for each species that do not have a strong temperature response?

*See response at line 125.*

9. l.153 DJF Maybe point to Table 1 as justification – but include actual months of sensitivity in Table 1 – see below for comment.

*Agreed.*

10. 165 – 169. Presumably for DJF so comparable with the tree-ring reconstruction

*Yes, it can be clarified.*

Results

11. 173 – 175 rewrite this a bit to be as clear as possible.

*A proposed change to:*

*Superposed epoch analysis was used to analyse the volcanic response of the eight tree species*
*to the 13 volcanic eruptions with SAOD > 0.08 between 1400 and 1990 CE (Figure 3). Two*
*composite responses are shown for each species; the response averaged across all sites ('All*
*chronology composite'), and the response calculated only from the site chronologies which*
*individually showed either a significant (positive or negative) response to volcanic eruptions*
*('Sensitive chronology composite').*

12. 178 Which species had a neutral response? List here.

*Agreed.*

13. 180 – 195 Include references to lagged responses shown in Figure…compare with later
comments on the 'lack of memory' in the NZ trees compared to the NH trees. Maybe also better
link this with the nature of the temperature responses (Figure S2-6).

*See response at line 180.*

Section 3.2

14. I am tempted to suggest that this could potentially go in Supplementary material (or even be
omitted altogether) to simplify the paper and amplify the main points of the paper. I think it is
useful to look at this, but not a key point. Also, the discussion here is a little confusing. In some
cases it is the difference between the species at the various sites that is noted but not really
described fully, but in other places, both species record a negative response. It would be useful
to discuss both the nature of the responses of the individual species at these sites and then if
they differ from the other species.

*We could agree to move Section 3.2 to Supplementary material; however, we think the*
*opportunity to compare different species at the same site directly is an interesting and valuable*
*addition to the paper and should not be omitted. It could be clarified that at some locations, the*
*climate response is not sufficient to result in a response in either species, but at other sites we*
*can observe differences between species.*

Section 3.3

15. 218 It would be good to preface this section with some statement about why focus on cedar
(why not do similar analyses for other species? – i.e. be explicit). One suggestion…begin with
a comment about how the cedar average showed a generally muted response, but this masks
very different individual site responses…and hence why this section of the results is useful.
While it is certainly understandable that the authors wish to consider this material in the main
manuscript, it may be worth considering whether at least some of this information could go in
the Supplementary? (so as not to distract too much from the bigger messages in the paper).

*With Section 3.2 moved to Supplementary and the addition of an introductory sentence/s*
*explaining why NZ cedar was selected for analysis that references Supplementary, we think the*
*results of Section 3.3 are important to be included in the main paper.*

16. 279 models – not CMIP models? Just be clear which models (reconstruction model or CMIP)
is being referred to.

Ok.

17. 284 – 287 So there isn't a substantial issue with memory in the temperature reconstruction, but there are lags in the species-level responses. Be careful how this is discussed throughout the Results and Discussion.

This can be discussed further.

Discussion.

18. As mentioned earlier, there is considerable repetition here (representation of Results) that clouds the text somewhat.

We can revise the discussion to remove the repetition of results.

19. Section 4.1 – Compare with the above. Why doesn't this play out in the temperature reconstruction? Would be good to comment on.

See response at line 180.

20. 340 – 345 This seems a little confused. Why separate silver beech from the other stress tolerators in this section? This section could probably be tightened up a bit.

In this paragraph, we compared the responses of the two beech species, as distinct from the remaining species, which are all conifers. We found it interesting that the beech species showed such different response characteristics. However, this discussion can be incorporated into the previous paragraphs if that is clearer for the reader.

21. 350 – 354. So how does this relate to strong responses in years 0 and 3?

We propose that the initial response in year 0 is foliage production, and in year 3, a secondary response is triggered by the cooler than average summer temperatures in year 0. Whether this response is due to a quasi-biennial flowering cycle or mast seeding or is related to normal cladode lifespan is a hypothesis. We favour the cladode senesce hypothesis due to our own, albeit limited, field observations.

22. 369 "…around 1000mm" This still seems relatively high, but how does it relate to the needs and distributional range (with respect to precipitation) of the species?

This section could be clarified by discussing soil moisture balance rather than precipitation. In average years, there is a summer soil moisture deficit at Takapari because evapotranspiration exceeds precipitation, whereas in Westland, there is almost always a precipitation surplus (see Fig. 3). We therefore hypothesise that we see a positive summer moisture response for the Takapari chronologies but not the Westland chronologies because they do not experience summer moisture stress.

[Figure]

**Figure 3 – Average monthly soil moisture deficit (mm) for a soil moisture capacity of 150 mm at two selected gauges: Hokitika, Westland, and Palmerston North, Manawatu-Wanganui. (Data sourced from: Chappell, 2015; Macara, 2016).**

23. 370 – 373. Again, mention of possible link to role of moisture. This seems to suggest that perhaps moisture-related variability should have been considered in this study? However, the low number of samples is of some concern, and perhaps this section should be shortened accordingly.

As New Zealand precipitation is very spatially variable, this analysis would need to be undertaken on a site-by-site basis. However, the locations of precipitation gauges are often not representative of conditions at the chronology sites due to distance (sometimes > 100 km), differences in altitude, orographic effects of the ranges, etc.

24. 392 I'm not sure this is counter-intuitive response given the negative response to temperature (Supplementary). L. 397-99. This mention of seasonality of growth again makes me wonder if this should have been more fully described for all species much earlier (the longer climate response window of pink pine for eg is interesting)?

We think the combination of cool and dry spring/summer conditions is counter-intuitive rather than just the temperature response, but we can revise this wording.

We also believe the different seasonality of growth has been addressed sufficiently in Tables 1 and 2 and Figures S3-S9.

25. Section 4.2 Needs considerable tightening up. Reference to moisture-related responses is speculative (but not unreasonable), but it needs to be couched that way. Also, while the results and discussion for cedar in Section 3.3 are suggestive, I don't think they should be presented as being THE causes of differences. They may well be, but further work, and closer examination across all the species would provide more evidence for this. At l. 433, other factors are mentioned. This reference should perhaps come earlier in this section to better set up the

discussion around the evidence presented in Section 3.3. Especially when the authors go on to discuss location in the landscape (l. 440 – 446). This isn't discussed in relation to the PCA results in Section 3.3. If these factors are so important, they should be mentioned in that section – do the PCA results reflect this?

The section can be revised. Please also see the response to RC2 for additional evidence for the findings of this section.

26. 422 " …sensitivity to temperature, including volcanic cooling…" So why include sites/chronologies not sensitive to temperature (or not at their limits)? This gets back to the locations of many of the SH tree-ring chronologies, and the relative lack of 'choice' compared to the NH.

See response at line 125.

27. 450 "..large on site-related…". Only Elevation and latitude really mentioned.

The wording can be revised.

28. 450 – 452. While it is great that the authors produced this new reconstruction and tested it for volcanic response, I have two points to make about it:

a. The main features of this (these) new reconstruction are not really described in the study. How does it differ from earlier reconstructions? (Obviously the climate target may differ, but does it show similar features? If not/if so, where….?). Is it different enough to be the potential reason for volcanic dimming being detectable here but not in previous reconstructions?

Here we present a comparison of our reconstructions to three previously published temperature reconstructions from New Zealand (Fig. 4). The Cook et al. (2002) and Palmer & Xiong (2004) reconstructions were downloaded from the NOAA paleoclimate database, and the authors provided the Duncan et al. (2010) reconstruction. All reconstructions show approximately consistent high and low-temperature periods and an increase in average temperature since ~1950.

[Figure]

**Figure 4 - New Zealand summer temperature reconstructions. a, b) DJF New Zealand average temperatures, this study; c) January-March temperature at Hokitika, Westland (Cook et al., 2002); d) Annual average New Zealand temperature (Duncan et al., 2010); e) February-March average New Zealand temperature (Palmer & Xiong, 2004). Reconstructed temperature is shown in grey, and the 20-year running mean is in black.**

Despite different climate targets and seasons, and large differences in the number and geographical distribution of the predictors, all reconstructions are significantly correlated (p <<

0.001). The highest correlation is between our reconstruction NZall and the pink pine reconstruction of NZ annual average temperatures by Duncan et al. (r = 0.62). Should the paper be recommended for publication, we propose briefly describing the reconstructions' main features and adding the figure above to Supplementary.

b. If previous reconstructions were compared with the regional dimming index in the same manner would the same result be produced? Has it been the compilation of volcanic eruptions based on their magnitude which has been the problem in the past? Or is it the combination of chronologies used? The target season? The season of the eruptions vs seasonality of tree growth without due attention to regional and global circulation patterns?

None of the studies discussed above considered a volcanic response, nor did any of the other published NZ temperature reconstructions for which the data are not available. Thus, we suspect the main reason volcanic responses have not been previously identified in New Zealand tree-ring temperature reconstructions is that no analysis has been done until now.

Palmer & Ogden (1992) and Norton (1992) are the only studies to our knowledge that looked for volcanic signals in tree rings from New Zealand. Both studies consider only the Tambora eruption of 1815, and their methodologies simply identify whether there were narrow rings in the years following the eruption event. Both conclude that the evidence is not sufficient to identify a volcanic signal. Our analysis and the results of Palmer et al. compare quite well: cedar chronologies declined for several years following an eruption, similar to the lagged and persistent pattern we observed over 13 eruptions, and the two years following the eruption show increased average ring width in their kauri chronologies. Similarly, our results for the two beech species correspond well with the results of Norton. The issue faced by these researchers is that they were seeking common responses across species, whereas our analysis, with the benefit of much more data, shows responses vary widely between species.

Without undertaking a detailed analysis, the global investigation of Krakauer & Randerson (2003) likely suffers from the compositing of the many different chronologies, which results in some of the volcanic signal cancelling as noise. This is the reason we caution against compositing too widely in our conclusions.

Conclusions

29. I still think the big news is that a volcanic signal was identified. The 'next big news' is related to the species-level responses.

These points can be rearranged.

30. 488 "..proxy selection…". This almost sounds like a choice amongst corals, trees, speleothems etc. Do you mean site selection?

Perhaps 'chronology' is better here.

31. 492 "….plant life history traits…". What is meant by this? Not really discussed in the manuscript. Maybe just be explicit to avoid confusion.

It can be replaced explicitly with stress tolerator/fast responder.

32. 500 "We found that…." Not convinced this is a major finding when it depends heavily on Section 3 and then later observations not related to the analysis in Section 3.3.

Ok.

33. 503 " …summer moisture…" I think this is too speculative to include in the conclusions.

It can be removed from the conclusions.

34. 513-515. But this study seemed to indicate that it didn't matter so much whether a subset of the most sensitive sites was used or not (especially for the temperature reconstruction). Reword.

Ok.

35. 518 – 521. Yes, this applies to other types of reconstructions as well. Note that several large databases include composites that are then used by modellers who may not appreciate these types of nuances.

Ok.

Figures and Tables

36. Figure 2. The orange and red are quite close to one another.  Might it be useful to darken the red so there is a clearer visual difference between the two?
Agreed

37. Figure 5 – colours for G4 and G5 difficult to tell apart for some.  Change one of the colours.
Agreed

38. Figure 8 – is it possible to use different symbols for the different species?
Yes.

39. Table 1 Could this table be merged with Table 2 that simply summarises the nature of the response.  Maybe also note which response is stronger, prior or current season?
Yes, the Table 1 column 'reported climate sensitivity', which summaries the referenced publications, could be replaced with the calculated responses from Table 2.

40. Table 2 – I think this could be merged with Table 1, but in discussing Table careful references to Figures S2-6 should be made.
Agreed

Supplementary

41. Figures S3 and S4.  It is unclear why master chronologies are included just for these two
species.  Perhaps more could be made of the differences between 'master series' and individual series for all species in the main text?  It would actually be good to see master series for all species included here given that species wide averages have been used in the main manuscript (Figure 3).
The master chronologies provided in S3 and S4 are the published master chronologies
downloaded from the ITRDB and associated with the references in Table 1. The published master chronologies may differ from the average chronology across sites used in our analysis because they are based on a subset of the chronologies. Certainly, we can also plot the temperature correlation of the average chronology for each species (i.e., shown in Figure 3). The addition of the table of site metadata should resolve any confusion.

*Technical comments*

Abstract

42. 28 "The has…" This has.

Agreed

**Introduction**

43. 88 amongst rather than between

Agreed

44. 89 Should "proxy" be "site"?

Perhaps 'chronology' is better to capture both sites and species.

**Methods**

45. 98 should "species depth" be "sample depth"?

Possibly. The chronology was excluded because it is the only chronology from that species, not because the chronology itself had insufficient sample depth; thus we chose 'species depth'. We are not wedded to the current wording.

46. 133 "…two species.." Maybe insert 'different' between these words?

Agreed.

**Results**

47. 175 "averaged across…." All sites of a species, not just all sites. Ditto in relation to sensitive chronologies.

Agreed.

## Discussion

48. 438 "…altitudinal range.." Altitudinal limit?

Yes, it should be the altitudinal limit.

49. 478 MDX – should be MXD?

Yes, this is a typo.

**References used in this response**

Allen, K. J., Cook, E. R., Evans, R., Francey, R., Buckley, B. M., Palmer, J. G., Peterson, M. J., & Baker, P. J. (2018). Lack of cool, not warm, extremes distinguishes late 20th Century climate in 979-year Tasmanian summer temperature reconstruction. *Environmental Research Letters*, *13*(3). https://doi.org/10.1088/1748-9326/aaafd7

Büntgen, U., Allen, K., Anchukaitis, K. J., Arseneault, D., Boucher, É., Bräuning, A., Chatterjee, S., Cherubini, P., Churakova (Sidorova), O. V., Corona, C., Gennaretti, F., Grießinger, J., Guillet, S., Guiot, J., Gunnarson, B., Helama, S., Hochreuther, P., Hughes, M. K., Huybers, P., … Esper, J. (2021). The influence of decision-making in tree ring-based climate reconstructions. *Nature Communications*, *12*(1). https://doi.org/10.1038/s41467-021-23627-6

Chappell, P. R. (2015). The climate and weather of Manawatu-Wanganui. *NIWA Science and Technology Series*, *66*, 40. http://docs.niwa.co.nz/library/public/NIWAsts66.pdf

Cook, E. R., Bird, T., Peterson, M., Barbetti, M., Buckley, B., D'Arrigo, R., & Francey, R. (1992).

Climatic change over the last millennium in Tasmania reconstructed from tree-rings. *Holocene*, *2*(3), 205–217. https://doi.org/10.1177/095968369200200302

Cook, E. R., Palmer, J. G., Cook, B. I., Hogg, A., & D'Arrigo, R. D. (2002). A multi-millennial palaeoclimatic resource from Lagarostrobos colensoi tree-rings at Oroko Swamp, New Zealand. *Global and Planetary Change*, *33*(3–4), 209–220. https://doi.org/10.1016/S0921-

8181(02)00078-4

Duncan, R. P., Fenwick, P., Palmer, J. G., McGlone, M. S., & Turney, C. S. M. (2010). Non-uniform interhemispheric temperature trends over the past 550 years. *Climate Dynamics*, *35*(7), 1429–1438. https://doi.org/10.1007/s00382-010-0794-2

Krakauer, N. Y., & Randerson, J. T. (2003). Do volcanic eruptions enhance or diminish net primary production? Evidence from tree rings. *Global Biogeochemical Cycles*, *17*(4). https://doi.org/10.1029/2003gb002076

Macara, G. R. (2016). The Climate and Weather of West Coast. *NIWA Science and Technology Series*, *72*, 40. http://docs.niwa.co.nz/library/public/NIWAsts72.pdf

Norton, D. A. (1992). New Zealand Temperatures, 1800-30. In C. R. Harington (Ed.), *The Year*

*without a summer? : world climate in 1816* (pp. 516–520). Canadian Museum of Nature.

Palmer, J. G., & Ogden, J. (1992). Tree-Ring Chronologies from Endemic Australian and New Zealand Conifers 1800-30. In C. R. Harington (Ed.), *The Year without a summer? : world climate in 1816* (pp. 510–515). Canadian Museum of Nature.

Palmer, J. G., & Xiong, L. (2004). New Zealand climate over the last 500 years reconstructed from

Libocedrus bidwillii Hook. f. tree-ring chronologies. *Holocene*, *14*(2), 282–289. https://doi.org/10.1191/0959683604hl679rr

Toohey, M., & Sigl, M. (2017). Volcanic stratospheric sulfur injections and aerosol optical depth from 500BCE to 1900CE Matthew. *Earth Syst. Sci. Data*, *9*, 809–831. https://doi.org/10.1108/eb058541

---

## Author Comment (AC2)

**Overview**

This study by Higgins et al. investigates the Southern Hemisphere (New Zealand, to be specific) tree growth response to volcanic events. In contrast to previous studies that barely found any response, this study presents evidence of clearly identified responses. The authors conducted superposed epoch analysis (SEA) on tree-ring chronologies by species and by sites, and by groups of cedar chronologies as a further analysis. With these analyses, they found that the volcanic response of the New Zealand trees could be positive, negative, and neutral, and site-related factors appear to be more important than species. Then they built temperature reconstructions based on these tree-ring chronologies, on which they conducted SEA, comparing to that of the climate model simulations. The comparison shows agreement between simulations and reconstructions, indicating that the New Zealand trees are reliable volcanism recorders.

In my opinion, these exciting results are of importance and interest to the community, and can stimulate further studies on Southern Hemisphere trees. The manuscript is overall in good quality, with a clear structure, and analyses being thorough and to the point. I have only a few minor suggestions that I list below. Once those have been addressed, I recommend the work be accepted for publication.

Thank you for your review. Please find the responses to your specific comments in blue below.

**Specific comments**

L28: "the" is a typo.

Yes, thank you.

Figs. 3, S1: We still need the y-axis label for chronologies.

The y-axis label (Average ring-width anomaly) will be added

Fig. 5: Similar to Fig. 3, the y-axis label is missing. The x-axis label ("Years since event year") should be put under the two columns since we have a map at the upper-left corner that does not share such x-axis label.

The y-axis label (Average ring-width anomaly) will be added, and the x-axis label moved.

Figs. 6, S7, S8, S10: The x-axis label is missing.

The x-axis label will be added.

Figs. S2-S6: It seems that the legends are raw codes without any explanation in the caption, and it would be difficult to understand for people who's not familiar with these codes.

We propose that a table with site meta-data for each code is added to Supplementary Information (attached at the end of this document). The legend for Figures S2-S6 would then be updated to link to the table.

L285-287: The readers will wonder why it is the case, and a pointer to the specific discussion section is needed here.

Agreed.

Northern Hemisphere high altitude/latitude chronologies predominantly used to determine the climatic effects of volcanic eruptions contain higher biological persistence than the chronologies we have used. In terms of the discussion in this paper, we would classify these trees as extreme stress tolerators. To illustrate, we refer to Table 9 in Cropper & Fritts (1981) which compares the ring width characteristics of arctic trees to more temperate Northern Hemisphere trees. The average first order autocorrelation of the series used to develop our temperature reconstructions is 0.53 (0.149-0.869) with standard deviation 0.15 compared to artic sites with average 0.62 (0.15 – 0.93) and standard deviation of 0.13.

L382: It seems that Figure 3 is the one to refer to, instead of Figure 2.

Yes, this needs to be updated to Figure 3.

L458-460: A pointer to Figure 7 is needed here.

Agreed.

L478: "MDX" is a typo.

Yes, thank you.

**References used in this response**

Cropper, J. P., & Fritts, H. C. (1981). Tree-ring width chronologies from the North American Arctic. *Arctic and Alpine Research*, *13*(3), 245–260. https://doi.org/10.2307/1551032

**Supplementary table 1 – Meta data for all chronologies used in this study.**

| Site | Species | Start | End | Longitude | Latitude | Altitude (m asl) | ITRDB Code | Notes |
|------|---------|-------|-----|-----------|----------|------------------|------------|-------|
| 1CAS | AGAU | 1559 | 1982 | -36.88 | 174.53 | 180 | newz082 | Cascades |
| 1HID | AGAU | 1679 | 2002 | -36.20 | 175.43 | 220 | newz083 | Hidden Valley |
| 1HUI | AGAU | 1720 | 1981 | -36.97 | 174.57 | 274 | newz085 | Huia |
| 1HUP | AGAU | 1483 | 1997 | -36.82 | 174.50 | 90 | newz084 | Huapai |
| 1KAT | AGAU | 1698 | 1996 | -37.60 | 175.87 | 350 | newz091 | Katikati |
| 1KAW | AGAU | 1710 | 1996 | -37.92 | 174.92 | 80 | newz087 | Kawhia |
| 1KON | AGAU | 1770 | 1976 | -37.07 | 175.13 | 335 | newz008 | Konini Forks |
| 1LTB | AGAU | 1790 | 1981 | -36.20 | 175.13 | 274 | newz086 | Little Barrier Island |
| 1MAS | AGAU | 1269 | 1998 | -36.90 | 175.55 | 350 | newz088 | Manaia Sanctuary |
| 1MOE | AGAU | 1360 | 1980 | -36.53 | 175.55 | 630 | newz089 | Mount Moehau |
| 1MWL | AGAU | 1580 | 1981 | -37.22 | 175.03 | 350 | newz090 | Mount William |
| 1PBL | AGAU | 1675 | 1981 | -35.18 | 173.75 | 305 | newz078 | Puketi Bluff |
| 1PKF | AGAU | 1504 | 2002 | -35.27 | 173.73 | 290 | newz079 | Puketi Forest |
| 1TRO | AGAU | 1408 | 2002 | -35.72 | 173.65 | 175 | 2 | Trounson Kauri Park |
| 1WFD | AGAU | 1628 | 1903 | -35.65 | 173.57 | 180 | newz022 | Waipoua Forest |
| 1WWF | AGAU | 1462 | 2002 | -35.37 | 173.28 | 468 | newz081 | Warawara Plateau |
| 2BON | HABI | 1463 | 1999 | -43.08 | 170.65 | 850 | 1 | Mount Bonar |
| 2CCP | HABI | 1410 | 1998 | -42.72 | 171.57 | 970 | 1 | Camp Creek |

| Site | Species | Start | End | Longitude | Latitude | Altitude (m asl) | ITRDB Code | Notes |
|------|---------|-------|-----|-----------|----------|------------------|------------|-------|
| 2CRS | HABI | 1483 | 1999 | -42.28 | 171.38 | 900 | 1 | Croesus Track |
| 2DBY | HABI | 1457 | 2010 | -47.03 | 167.72 | 100 | newz118 | Doughboy - Adams Hill |
| 2ELD | HABI | 1338 | 1999 | -45.75 | 167.47 | 750 | 1 | Eldrig Peak |
| 2GLS | HABI | 1461 | 1999 | -41.62 | 172.03 | 950 | 1 | Mount Glasgow |
| 2HEL | HABI | 1407 | 2013 | -46.98 | 167.75 | 100 | newz119 | Hellfire Ruggedy Mt |
| 2MAP | HABI | 1567 | 1976 | -45.53 | 167.30 | 305 | newz010 | Manapouri Dam |
| 2MAT | HABI | 1508 | 1999 | -41.57 | 172.32 | 1060 | 1 | Matiri Range |
| 2MEL | HABI | 1440 | 1999 | -42.50 | 171.83 | 1050 | 1 | Mount Elliot |
| 2MGR | HABI | 1400 | 1999 | -42.95 | 170.82 | 865 | 1 | Mount Greenland |
| 2MTF | HABI | 1367 | 1999 | -42.67 | 171.33 | 750 | 1 | Mount French |
| 2OMO | HABI | 1578 | 1999 | -43.40 | 170.10 | 320 | 1 | Omoeroa Saddle |
| 2PEG | HABI | 1667 | 1991 | -46.92 | 167.73 | 450 | 2 | Pegasus Stewart Island |
| 2PUT | HABI | 1646 | 1993 | -40.67 | 175.52 | 650 | newz010 | Putara |
| 2SPD | HABI | 1447 | 1999 | -46.37 | 169.05 | 560 | 1 | Slopedown Hill |
| 2TKG | HABI | 1450 | 1999 | -42.65 | 171.50 | 950 | 1 | Mount Tekinga |
| 2TKP | HABI | 1708 | 1995 | -40.08 | 176.00 | 800 | NEWZ076 | Takapari |
| 2TOS | HABI | 1590 | 1998 | -42.98 | 170.85 | 210 | 1 | Totara Saddle |
| 3AHA | LACO | 1209 | 2000 | -42.38 | 171.80 | 244 | newz005 | Ahaura |
| 3FLG | LACO | 1230 | 2003 | -42.50 | 171.72 | 200 | newz120 | Flagstaff Creek |
| 3MWO | LACO | 1464 | 1976 | -39.35 | 175.48 | 1000 | newz011 | Mangawhero River Bridge |
| 3ORO | LACO | 470 | 1999 | -43.23 | 170.28 | 110 | newz121 | Oroko Swamp |
| 3SWF | LACO | 1130 | 1969 | -43.13 | 170.40 | 200 | newz122 | Saltwater Forest |
| 4AHA | LIBI | 1303 | 2009 | -42.38 | 171.80 | 244 | newz127 | Ahaura |
| 4ARM | LIBI | 1446 | 1958 | -43.83 | 173.00 | 731 | newz007 | Armstrong Reserve |
| 4CCC | LIBI | 1064 | 2010 | -42.72 | 171.57 | 965 | newz124 | Camp Creek |
| 4CLW | LIBI | 1450 | 1991 | -39.63 | 176.10 | 1220 | newz064 | Clearwater |
| 4CRG | LIBI | 1492 | 2010 | -45.83 | 170.53 | 576 | newz128 | Mount Cargill |
| 4CRK | LIBI | 1460 | 1978 | -43.08 | 170.98 | 800 | newz039 | Cream Creek |
| 4EMT | LIBI | 1616 | 1990 | -39.25 | 174.08 | 1050 | newz003 | Mount Egmont |
| 4FLG | LIBI | 1464 | 2004 | -42.50 | 171.72 | 200 | newz125 | Flagstaff Creek |
| 4FLH | LIBI | 1683 | 1991 | -41.27 | 172.60 | 950 | newz065 | Flanagans Hut |
| 4HIT | LIBI | 1431 | 1991 | -39.53 | 175.73 | 976 | newz066 | Hihitahi |
| 4MOA | LIBI | 1490 | 1991 | -40.93 | 172.93 | 1036 | newz067 | Moa Park |
| 4MTF | LIBI | 1330 | 1999 | -42.67 | 171.33 | 855 | newz126 | Mount French |
| 4MWO | LIBI | 1662 | 1976 | -39.35 | 175.48 | 1000 | newz012 | Mangawhero River Bridge |
| 4NET | LIBI | 1625 | 1990 | -39.28 | 174.10 | 991 | newz014 | North Egmont |
| 4OHT | LIBI | 1585 | 1991 | -39.62 | 176.12 | 1140 | newz068 | Ohutu Ridge |
| 4OKA | LIBI | 1732 | 1976 | -46.38 | 169.45 | 305 | newz016 | Owaka |
| 4RAH | LIBI | 1480 | 2012 | -42.32 | 172.12 | 672 | newz129 | Rahu Saddle |
| 4RUC | LIBI | 1473 | 1991 | -39.63 | 176.18 | 1200 | newz069 | Ruahine Corner |
| 4STR | LIBI | 1626 | 1990 | -39.32 | 174.12 | 860 | newz071 | Stratford side - East Egmont |

| Site | Species | Start | End | Longitude | Latitude | Altitude (m asl) | ITRDB Code | Notes |
|------|---------|-------|-----|-----------|----------|------------------|------------|-------|
| 4TKP | LIBI | 1256 | 1992 | -40.07 | 175.98 | 838 | newz062 | Takapari Road |
| 4TOA | LIBI | 1511 | 1992 | -39.23 | 175.43 | 1160 | newz072 | Hauhangatahi Site A |
| 4TOB | LIBI | 1332 | 1992 | -39.23 | 175.43 | 1100 | newz073 | Hauhangatahi Site B |
| 4TOC | LIBI | 1213 | 1992 | -39.23 | 175.43 | 1000 | newz074 | Hauhangatahi Site C |
| 4TRK | LIBI | 1526 | 1978 | -43.08 | 170.97 | 925 | newz055 | Tarkus Knob |
| 4UWR | LIBI | 1140 | 1992 | -38.68 | 177.20 | 854 | newz063 | Urewera |
| 4WBF | LIBI | 1674 | 1992 | -43.07 | 171.28 | 780 | newz075 | Wilberforce |
| 5BOR | NOME | 1389 | 2007 | -45.78 | 167.37 | 200 | 2 | Borland |
| 5KEA | NOME | 1580 | 1980 | -43.87 | 169.78 | 1150 | newz036 | Kea Flat |
| 5LKE | NOME | 1676 | 1980 | -45.25 | 167.48 | 950 | newz048 | Lake Eyles |
| 5LKO | NOME | 1584 | 1980 | -45.30 | 167.68 | 1000 | newz051 | Lake Orbell |
| 5UHV | NOME | 1710 | 1980 | -44.77 | 168.00 | 950 | newz033 | Upper Hollyford Valley |
| 5UTV | NOME | 1622 | 1979 | -45.20 | 167.65 | 1000 | newz054 | Upper Takahe Valley |
| 6GHC | NOSO | 1795 | 2006 | -43.25 | 171.75 | 870 | newz046 | Ghost Creek |
| 6HDC | NOSO | 1730 | 1979 | -43.13 | 171.60 | 1350 | newz037 | Hidden Creek |
| 6LCV | NOSO | 1730 | 1979 | -43.08 | 171.72 | 1350 | newz035 | Lower Cass Valley |
| 6LGH | NOSO | 1740 | 1979 | -43.08 | 171.70 | 1400 | newz031 | Logos Hill |
| 6LGS | NOSO | 1760 | 1979 | -43.05 | 171.60 | 1300 | newz024 | Lagoon Saddle |
| 6LKP | NOSO | 1713 | 2006 | -43.12 | 171.78 | 970 | newz049 | Lake Pearson |
| 6MKW | NOME | 1730 | 1979 | -43.05 | 171.68 | 1275 | newz023 | Mirkwood |
| 6RTC | NOSO | 1787 | 2006 | -43.15 | 171.80 | 950 | newz052 | Rata Creek |
| 6SSS | NOSO | 1760 | 1979 | -43.05 | 171.72 | 1250 | newz030 | Snowslide Stream |
| 6TKV | NOSO | 1630 | 1979 | -45.30 | 167.68 | 1100 | newz031 | Takahe Valley |
| 6TST | NOSO | 1840 | 1979 | -45.28 | 167.65 | 1000 | newz032 | Takahe Stream |
| 6WND | NOSO | 1760 | 2006 | -43.08 | 171.58 | 1350 | newz053 | Windy Creek |
| 7PLC | PHAL | 1717 | 2015 | -42.90 | 171.57 | 915 | newz130 | Pegleg Creek |
| 8WER | PHGL | 1740 | 1976 | -38.57 | 175.70 | 518 | newz020 | Waimanoa Ecological Reserve |
| 8WHS | PHGL | 1550 | 1986 | -38.65 | 175.63 | 780 | newz056 | Waihora Stream |
| 8WKT | PHGL | 1535 | 1976 | -38.70 | 177.20 | 853 | newz009 | Lake Waikareiti |
| 8WPA | PHGL | 1585 | 1976 | -35.68 | 173.55 | 244 | newz022 | Waipoua Forest |
| 9OWI | PHTR | 1709 | 1976 | -41.12 | 173.67 | 15 | newz015 | Okiwi |
| 9PAP | PHTR | 1779 | 1975 | -36.12 | 174.25 | 160 | newz001 | Paparoa |
| 9WHH | PHTR | 1613 | 1986 | -38.70 | 175.60 | 575 | newz058 | Waihaha Terrace |
| 9WHL | PHTR | 1650 | 1985 | -38.65 | 175.67 | 640 | newz057 | Waihora Lagoon |
| 9WMU | PHTR | 1664 | 1976 | -37.03 | 175.53 | 61 | newz021 | Waiomu |
| 1Kauri | AGAU | 0 | 2002 | -36.20 | 175.43 | na | 2 | Kauri network |
| 2Pink | HABI | 1400 | 1999 | -43.23 | 170.28 | na | 2 | Pink pine network |
| 3Silver | LACO | 0 | 2003 | -43.23 | 170.28 | na | 2 | South Island silver pine |

1 https://researcharchive.lincoln.ac.nz/handle/10182/2141

2 Private collection

---

## Author Comment (AC3)

Response by authors to Reviewer Comment 2

**General Comments**

Higgins et al. show that New Zealand tree rings can indeed record past volcanic events. They effectively address the research questions they set out to answer. They find that the nature of the response to volcanic dimming varies across species, categorizing species as either "fast responders" of "stress tolerant." With this mixed response between species, they find that site-related factors are more important to the displayed volcanic response in tree-ring width. They additional develop two austral summer temperature reconstructions for New Zealand, which show evidence of cooling from past volcanic events. The response to past volcanic eruptions in these reconstructions shows good agreement with climate model temperature anomalies following volcanic eruptions. The authors competently shows that New Zealand tree-ring width is a reliable regional indicator of volcanic climate response. They add further nuance however and underline the importance of species/site selection, which will be very useful for future studies in this region that wish to optimize sample selection. I believe this publication is fit for publication after minor revisions and will be useful to the research community.

Thank you for your review. Please find the responses to your specific comments in blue below.

**Specific Comments**

In general I think you need to be more specific with the use of the term "dimming". I'm assuming you're using this term to refer to the increase in SAOD but this should be clearly stated to avoid confusion. The "dimming" term is used throughout the text as a catch all for the effects that could affect tree-ring width, but add specificity where you can. There also needs to be more discussion on the effect of light availability changes, or dimming, and how it could effect final tree-ring width. Particularly in your discussion of the kauri growth benefit (line 393-394). Line 62-63 is another part of the text with opportunity to add more discussion on effects of radiation changes from volcanic eruptions. Here are some references you could use to expand this discussion:

Robock, A. (2005). Cooling following large volcanic eruptions corrected for the effect of diffuse radiation on tree rings. Geophysical Research Letters, 32(6). https://doi.org/10.1029/2004gl022116
Tingley, M. P., Stine, A. R., & Huybers, P. (2014). Temperature reconstructions from tree- ring densities overestimate volcanic cooling. Geophysical Research Letters, 41(22), 7838– 7845.

Thank you for the comment and additional references. In terms of dimming, we will clarify wherever possible whether we are referring specifically to the increase in SAOD or to cooler temperatures because of dimming.

Our assumption that kauri receives a growth benefit from decreased evaporative demand following volcanic events is due to previous studies using dendrometer bands and the results of this study. Fowler et al. (2005) show that kauri growth rates are greatest over the austral spring (Sept-Nov), declining steeply over the summer months when evapotranspiration exceeds precipitation. Their results are not entirely consistent with an earlier study by Palmer & Ogden (1983), which showed peak growth continuing until the mid-summer before declining steeply. However, the sites included in Palmer & Ogden were at a higher altitude (245–720 m) than the site in Fowler et al., which could explain the delay in timing. Critical to our moisture stress hypothesis, Palmer & Ogden did not see a summer cessation of growth in their highest altitude site, Mt Moehau, which receives moisture from condensation and fog drip as well as rainfall. This additional information should be included in the discussion to support our assertions.

Nevertheless, we agree with the reviewer that our discussion neglects the potential benefit of light availability changes, which should also be included.

Line 109-114 How robust is this event list? Is there a secondary dataset you could use to test? Would you get the same events with the same SAOD thresholds? If there isn't a comparable dataset, I'm not

too concerned with this, but I think the choice of this dataset over potential others needs to be explained if it can change the final list of events used.

Yes, other datasets exist, for example, Crowley & Unterman (2012) and Gao et al. (2008). We used Toohey & Sigl (2017) as it is the most recent compilation of ice core data and has been used in other tree-ring studies of volcanic impacts, e.g., (Rao et al., 2019; Zhu et al., 2020). Here, we compare the event selection from our study with the SAOD estimates of Crowley & Unterman. We have not used Gao et al. for comparison because their spatially resolved dataset is provided as stratospheric loading and needs to be converted to SAOD. Selecting suitable conversion parameters for this dataset is beyond our expertise.

Crowley & Unterman provide their estimates of SAOD in two latitudinal bands for the Southern Hemisphere (0-30°S and 30-90°S), and thus we cannot select the same regional threshold (30-50°S) used in the main paper. Instead, we have chosen the Southern Hemisphere average across the two bands, as this was the most consistent with our original threshold. As Table 1 shows, event selection between the two datasets is largely consistent. Potential reasons for the differences, including the underlying ice core data and differences in methodology, are discussed in Toohey & Sigl (2017).

Table 1 - Comparison of event years selected from two ice core datasets.

| Eruption date (month/year) | Eruption | Toohey & Sigl (30-50°S) | Crowley & Unterman (0-90°S) |
|---|---|---|---|
| 1441 | Unknown | *Not selected* | > 0.04 |
| 1452 | Kuwae | > 0.04 | *Not selected* |
| 1457 | Unknown | > 0.08 | > 0.08 |
| 2/1477 | Bárðarbunga | > 0.04 | > 0.08 |
| 1588 | Unknown | *Not selected* | > 0.04 |
| 1595 | Unknown | > 0.08 | > 0.08 |
| 2/1600 | Huaynaputina | > 0.08 | > 0.08 |
| 1620 | Unknown | > 0.04 | > 0.04 |
| †12/1640 | Parker | > 0.08 | > 0.08 |
| 1653 | Unknown | > 0.04 | *Not selected* |
| 1673 | Gamnokara | > 0.04 | > 0.08 |
| 1694 | Unknown | > 0.08 | > 0.08 |
| 1761 | Unknown | > 0.04 | *Not selected* |
| 5/1783 | Grímsvötn Asama | > 0.08 | *Not selected* |
| 1804 | Unknown | *Not selected* | > 0.04 |
| 1809 | Unknown | > 0.08 | > 0.08 |
| 4/1815 | Tambora | > 0.08 | > 0.08 |
| 1831 | Babuyan Claro | > 0.04 | Not selected |
| 1/1835 | Cosigüina | > 0.08 | > 0.08 |
| †12/1861 | Makian | > 0.04 | > 0.04 |
| 8/1883 | Krakatau | > 0.08 | > 0.08 |
| 10/1902 | Santa Maria | not modelled | > 0.04 |
| 3/1963 | Agung | not modelled | > 0.08 |
| 3/1982 | El Chicon | not modelled | > 0.04 |

In Figure 1, we compare the two NZ temperature reconstructions using both datasets. There are some differences, as can be expected from averaging over a different subset of events, most notable a larger response to the Toohey & Sigl event list in b). There are also some issues with the compositing in c), with values in the normalisation period not close to 0, due to the small number of events in the Crowley & Unterman event list and noise in the NZall reconstruction. However, the results are unchanged; the NZ temperature reconstructions significantly respond to volcanic events in year t+1.

[Figure]

**Figure 1 - Comparison of SEA analysis using the two event years sets (Table 1) for the NZall and NZsens reconstructions.**

We believe that we have adequately accounted for the uncertainty around event selection by calculating confidence intervals as described in lines 162-165 of the main manuscript. We used bootstrapping to calculate the confidence intervals from 1000 replications of a subset of the events selected under the two SAOD thresholds. Note that as there are fewer events in Crowley & Unterman, the confidence intervals for eruptions with SAOD > 0.08 were constructed from only 200 bootstrap replications. This analysis provides an indication of how selecting a different event list could have affected the results.

Line 438-439 You need to support the statement that sites with high exposure to prevailing winds are more sensitive to low growing season temperatures, either from the literature or from your own analysis.

Line 444 Similar to the point above, you need to support this statement.

These points are made in reference to the conditions at the individual kauri sites, and this can be made clearer. However, evidence to support the lower sensitivity of sites experiencing mesic conditions and closed-canopy forests can be added with reference to Phipps (1982), and many previous studies support the difference in climate sensitivity due to aspect (e.g., Dang et al., 2007), especially when windward sites are exposed to prevailing winds (e.g., Rozas et al., 2013).

Figure 1 Add a legend for the elevation. This is important context for your conclusions as elevation is an important site characteristic.

An elevation legend can be added. Also, in response to RC1 and CC1, we have proposed the addition of a table of site metadata, which will allow readers to cross-reference the results with the coordinates and elevation data.

Figure 3 Caption "...the number of chronologies are shown in brackets/square brackets." Make it clear which bracket type refers to which chronology. Adding the word "respectively" will work.

Agreed.

**Technical**

Line 22 proxy --> site/species Using proxy sounds like you are expanding into non tree- ring proxies like coral for example.

Perhaps 'chronology' is best to capture both sites and species.

Line 46-49 Awkward sentence structure

This can be revised to improve readability, a proposed change to:

*There are several potential explanations for the considerable discrepancy between proxy reconstructions and climate models in the Southern Hemisphere. These include the underestimation of the moderating effects of the ocean on post-eruption cooling in climate models, changes to the hydrological cycle in response to volcanic cooling, uncertainties in volcanic forcing data, and/or proxy noise and spatial distribution (Neukom et al., 2018; Zhu et al., 2020).*

Line 51-52 tree-ring data

Agreed.

Line 89 proxy --> site/species

As per line 22, propose a change to 'chronology'.

Line 114-115 awkward sentence structure

This can be revised to improve readability, a proposed change to:

*The SAOD magnitude corresponding to a substantial temperature response is unknown before analysis. However, selecting a magnitude post-analysis based on the observed response risks biasing the results (Haurwitz and Brier, 1981). Therefore, two different SAOD thresholds were selected.*

Line 140 specify season

Agreed, DJF to be specified.

Line 436 add a call to Figure 5

Agreed.

Line 458-460 awkward sentence structure

This can be revised to improve readability, a proposed change to:

*We expected to find a substantially greater volcanic response in NZsens compared to NZall. However, while NZsens does show a larger post-volcanic temperature response, the overlap in the confidence intervals for both reconstructions indicates that the difference between their responses is not significant.*

Line 478 MDX-->MXD typo

Yes, thank you.

Line 482 Add call to Figure 7

Agreed.

Line 508 proxy --> species/site

Agreed.

**References used in this response**

Crowley, T. J., & Unterman, M. B. (2012). Technical details concerning development of a 1200-yr proxy index for global volcanism. *Earth System Science Data Discussions*, *5*(1), 1–28. https://doi.org/10.5194/essdd-5-1-2012

Dang, H., Jiang, M., Zhang, Q., & Zhang, Y. (2007). Growth responses of subalpine fir (Abies fargesii) to climate variability in the Qinling Mountain, China. *Forest Ecology and Management*, *240*(1–3), 143–150. https://doi.org/10.1016/j.foreco.2006.12.021

Fowler, A. M., Lorrey, A., & Crossley, P. (2005). Seasonal growth characteristics of kauri. *Tree-Ring Research*, *61*(1), 3–19. https://doi.org/10.3959/1536-1098-61.1.3

Gao, C., Robock, A., & Ammann, C. (2008). Volcanic forcing of climate over the past 1500 years: An improved ice core-based index for climate models. *Journal of Geophysical Research Atmospheres*, *113*(23). https://doi.org/10.1029/2008JD010239

Palmer, J. G., & Ogden, J. (1983). A dendrometer band study of the seasonal pattern of radial increment in kauri (agathis australis). *New Zealand Journal of Botany*, *21*(2), 121–125. https://doi.org/10.1080/0028825X.1983.10428535

Rao, M. P., Cook, E. R., Cook, B. I., Anchukaitis, K. J., D'Arrigo, R. D., Krusic, P. J., & LeGrande, A. N. (2019). A double bootstrap approach to Superposed Epoch Analysis to evaluate response uncertainty. *Dendrochronologia*, *55*(February), 119–124. https://doi.org/10.1016/j.dendro.2019.05.001

Rozas, V., García-González, I., Pérez-De-Lis, G., & Arévalo, J. R. (2013). Local and large-scale climatic factors controlling tree-ring growth of Pinus canariensis on an oceanic island. *Climate Research*, *56*(3), 197–207. https://doi.org/10.3354/cr01158

Toohey, M., & Sigl, M. (2017). Volcanic stratospheric sulfur injections and aerosol optical depth from 500BCE to 1900CE Matthew. *Earth Syst. Sci. Data*, *9*, 809–831. https://doi.org/10.1108/eb058541

Zhu, F., Emile-Geay, J., Hakim, G. J., King, J., & Anchukaitis, K. J. (2020). Resolving the Differences in the Simulated and Reconstructed Temperature Response to Volcanism. *Geophysical Research Letters*, *47*(8), 1–12. https://doi.org/10.1029/2019GL086908

---

## Author Response (AR1)

Dear Dr Sigl,

Re: Revision of manuscript reference cp-2021-171

Please find attached a revised version of our manuscript originally entitled "*Do Southern Hemisphere tree rings record past volcanic signals?*" which we would like to resubmit for consideration for publication in *Climate of the Past*.

The three reviewer's comments were highly insightful and enabled us to improve the quality of our manuscript. In the following pages are our point-by-point responses to each of the comments, and to the additional editorial points you raised. A table summarising all changes is presented first, followed by the full responses.

Under your recommendation, the title was revised to "*Do Southern Hemisphere tree rings record past volcanic signals? A case study from New Zealand*". Following the reviewer's comments, additional analysis of the two New Zealand temperature reconstructions and a comparison to previously published reconstructions were added. A comparison to a second volcanic event dataset was also incorporated. Finally, we have considerably reworked the results and discussion sections to a) clarify the terminology of volcanic radiation changes, b) strengthen the links between species' temperature sensitivities and volcanic responses, and c) remove repetition and streamline the discussion.

We thank you for your time and consideration of our manuscript.

Yours sincerely,

Philippa Higgins

**Line by line response to Editor's comments**

| Section/Figure | Summary of comment | Action – reference to revised manuscript |
|---|---|---|
| Title | Change Southern Hemisphere to New Zealand | Title changed to '*Do Southern Hemisphere tree rings record past volcanic events? A case study from New Zealand*' |
| Line 16 | Change events to eruptions | Changed. |
| Line 32 | Changes Iles (2013) reference to original paper

Reference to Robock (2000) not accurate | Reference updated

Robock (2000) substituted with Robock (2005). |
| Line 34 | Supplement Tejedor reference with pure proxy reference | References to Wilson et al., (2016) and D'Arrigo et al., (2013) added. |
| Line 39 | Have any studies from South America been published? | We found one study Villalba and Boninsegna (1992) from South Amercia. This reference is discussed at Line 41-42. |
| Line 44/45 | Please specify what specific signal model simulations showed. | Clarification (reduced mean surface air temperatures) added Line 49. |
| Line 95 | Citations missing | Missing citations added to bibliography, all citations checked, and bibliography formatting updated to meet journal requirements. |
| Figure 2 | Fix typo and remove reference to Kuwae eruptions as sufficient evidence of location not available | Figure 2 updated |
| Table 1 | Have any new tree-ring chronologies been constructed since 2006? | No. Some sites have been updated since then (e.g., Pegleg Creek in 2015) but there have been no new sites developed other than sub fossil kauri records which we don't consider in this analysis.

A discussion of the timing of chronology development versus the availability of global aerosol loading reconstructions has been included in the comparison to previous studies at Line 489-500. |

**Line by line response to Reviewer 1, detailed response page 5**

| Section/Figure | Summary of comment | Action – reference to revised manuscript |
|---|---|---|
| Figs. 3, S1 | Add y-axis | Added, see revised Fig. 3 Line 199

See Fig. S7 in Supplemental |
| Fig. 5 | Add y-axis and reposition x-axis | Actioned, see new Fig. 4 Line 253 |
| Figs. 6, S7, S8, S10 | Add x-axis | Added, see new Fig. 5 Line 273

See Figs. S9, S10, S14 in Supplemental |
| Figs. S2-S6 | Add legend for chronology codes | New Table S1 added with meta data for all chronologies, see Supplemental |
| L285-287 | Explain why there is less persistence in New Zealand trees compared to Northern Hemisphere trees | Explanation added at Lines 516-524 |
| Technical | Typos | Addressed. |

**Line by line response to Reviewer 2, detailed response page 7**

| Section/Figure | Summary of comment | Action |
|---|---|---|
| General | Be more specific with the use of the term "dimming". | Clarification provided where possible between increase in SAOD and decreased temperatures, e.g., Lines 21, 68, 324, 343 |
| General | Add more discussion on the effect of light availability changes to final ring widths | Discussion added at Lines 62-63, 361-364, 396-403, 454-457 with reference to Tingley et al., (2014); Gu et al., (2003); Zweifel et al., (2021); Fatichi et al., (2019). |
| Line 109-114 | Is there a secondary dataset to test in addition to Toohey & Sigl (2017). | SEA analysis of the temperature reconstructions repeated with the dataset from Crowley & Unterman (2013).

Results added in Table S3 and Fig. S12 in Supplemental.

Discussion of results added to main manuscript at Lines 482-489. |
| Line 438-439 | Support statement that sites with high exposure to prevailing winds are more sensitive to low growing season temperatures | Additional analysis added at Lines 458-467. |
| Line 444 | Support statement that sites with mesic conditions are less sensitive to low growing season temperatures | Reference to Phipps (1982) added at Line 453-454. |
| Figure 1 | Add a legend for the elevation | Added, see revised Fig. 1 at Line 107. |
| Figure 3 | Clarify brackets used in caption | Clarified, see Line 201. |
| Technical | Typos, awkward wording | Addressed Line 51-54, Line 132-134, and Line 470-472. |

**Line by line response to Community Comment 1, detailed response page 11**

| Section/Figure | Summary of comment | Action |
|---|---|---|
| General | Restructure information to high result of temperature reconstruction | Not actioned. We believe the paper structure is appropriate in its current form. |
| General | Why did this study find evidence for volcanic impacts when previous studies did not. | A comparison to previous studies with explanation of the difference to our study added at Lines 490-501. |
| General | Include more details of the study sites | New Table S1 added to Supplementary. |
| General | Better links between Figs. S2-S6 showing species' temperature sensitivity and the results. | Considerable reworking of text to highlight these links at Line 218-232, and Line 421-428.

Addition of new Fig. S15 in Supplementary. |
| General | Highlight that elevation-moisture stress link is a hypothesis. | Additional explanation added and language clarified Lines 434-440. |
| General | Address potential discrepancy between lagged volcanic signals in tree rings but not in temperature reconstruction. | Addressed Lines 516-524. |
| Introduction - 1 | Might seasonality of signal matter | Discussion added at Line 371-373. |

| | | |
|---|---|---|
| Introduction - 2 | Regarding high altitude/latitude sites | Not actioned, we believe the text is appropriate in its current form. |
| Introduction - 3 | Reference to hydroclimate | Not actioned as comment unclear. |
| Introduction - 4 | Awkward sentence | Revised Line 80-82. |
| Methods - 5 | Revise line 104 for clarity | Revised Line 124. |
| Methods - 6 | Possible error in number of events | Clarified Line 138-139. |
| Methods – 7 | Event selection | Not actioned as we believe text is clear. |
| Methods – 8 | Potential pre-screen non sensitive chronologies before analysis. | Not actioned as making this change would have negligible impact on the results – see full response. |
| Methods – 9 | Include actual months of sensitivity to Table 1. | Added in revised Table 1. |
| Methods - 10 | Clarify season line 165. | DJF added to Line 186. |
| Results – 11 | Clarify lines 173-175 | Clarified Line 193-196. |
| Results – 12 | Add species with a neutral response | Added Line 207. |
| Results - 13 | No lagged response in temperature reconstruction | Reference to later discussion added Line 300. Discussion added Lines 520-524. |
| Section 3.2 - 14 | Remove section. | Section removed. Fig. 4 moved to Supplemental as Fig. S8. |
| Section 3.3 - 15 | Add explanation for focus on cedar | Added Line 234-236. |
| Section 3.3 - 16 | Clarify which models line 279. | Clarified Line 292. |
| Section 3.3 - 17 | Address potential discrepancy between lagged volcanic signals in tree rings but not in temperature reconstruction. | Addressed Lines 520-524. |
| Discussion - 18 | Repetition of results | Considerable reworking of Discussion with repetitive text deleted. |
| Discussion - 19 | Address potential discrepancy between lagged volcanic signals in tree rings but not in temperature reconstruction. | Addressed Lines 520-524. |
| Discussion - 20 | Tighten discussion lines 340-345 | Paragraph reordered Line 333-352. |
| Discussion - 21 | Improved explanation of toatoa response line 350 | Explanation revised Line 353-364. |
| Discussion – 22/23 | Shorten section due to low number of samples. | Due to revisions including the removal of Section 2.3, this discussion was deleted. |
| Discussion - 24 | Revise wording around 'counter-intuitive climate response' Describe all species climate sensitivities earlier in article. | Wording revised Line 387. Summary of climate sensitivities added to Section 2.1 and Table 1. |
| Discussion - 25 | Section 4.2 needs tightening up | Considerable reworking of Discussion to address multiple comments resulted in Section 4.2 being deleted. Some text from Section 4.2 has been incorporated into Section 4.1 at Lines 365-379. |

| | | |
|---|---|---|
| Discussion - 26 | Why were sites/chronologies not sensitive to temperature (or not at their limits) included. | Discussion added at Line 422-428. |
| Discussion - 27 | 450 "..large on site-related...". Only Elevation and latitude really mentioned. | Considerable reworking of Section 4.3 to discuss other site-related factors. |
| Discussion - 28 | What are the main features of the temperature reconstructions and how to they compare to previous reconstructions?

Why didn't previous studies identify volcanic signals in their reconstructions? | Discussion of key features and comparison to previous studies added at Line 279-285 and addition of Fig. S11 and Table S5 to Supplemental.

Explanation added Line 490-501. |
| Conclusions – 29 | Temperature response should come first in conclusion. | Not actioned as per response to the comment in the first line of this table. |
| Conclusions – 30 | Minor wording | Revised. |
| Conclusions - 31 | What is meant by plant life history traits | Revised discussion is Section 4.1 should clarify this, so no change to conclusion. |
| Conclusions – 32/33 | Findings too speculative for conclusions | Text has been revised Line 539-547. |
| Conclusions - 34 | Line 513-515 not supported by results. Reword. | Text removed. |
| Conclusions - 35 | Comment regarding broader applicability of findings. | Comment - action not required. |
| Figs/Tables - 36 | Figure 2 – colour scheme | Colours adjusted in revised Fig. 2. |
| Figs/Tables - 37 | Figure 5 – colour scheme | Colours adjusted in revised Fig. 4. |
| Figs/Tables - 38 | Figure 8 – add symbols for species | Symbols adjusted in revised Fig. 7. |
| Figs/Tables – 39/40 | Merge Tables 1 and 2 | Tables 1 and 2 merged.

Additional references to Table 1 and Figs. S1-S6 added at Line 111-122. |
| Supplementary | Why are master chronologies not included in all temperature sensitivity figures? | Considerable reworking of Figs. S1-S6 to address this comment, plus addition of Table S1 in Supplementary. |
| Technical comments | Minor wording, typos | Addressed. |

**Detailed response to Reviewer 1**

**Overview**

This study by Higgins et al. investigates the Southern Hemisphere (New Zealand, to be specific) tree growth response to volcanic events. In contrast to previous studies that barely found any response, this study presents evidence of clearly identified responses. The authors conducted superposed epoch analysis (SEA) on tree-ring chronologies by species and by sites, and by groups of cedar chronologies as a further analysis. With these analyses, they found that the volcanic response of the New Zealand trees could be positive, negative, and neutral, and site-related factors appear to be more important than species. Then they built temperature reconstructions based on these tree-ring chronologies, on which they conducted SEA, comparing to that of the climate model simulations. The comparison shows agreement between simulations and reconstructions, indicating that the New Zealand trees are reliable volcanism recorders.

In my opinion, these exciting results are of importance and interest to the community and can stimulate further studies on Southern Hemisphere trees. The manuscript is overall in good quality, with a clear structure, and analyses being thorough and to the point. I have only a few minor suggestions that I list below. Once those have been addressed, I recommend the work be accepted for publication.

**Specific comments**

L28: "the" is a typo.

Addressed Line 31

Figs. 3, S1: We still need the y-axis label for chronologies.

Added, see revised Fig. 3 Line 199, Fig. S7 in Supplemental

Fig. 5: Similar to Fig. 3, the y-axis label is missing. The x-axis label ("Years since event year") should be put under the two columns since we have a map at the upper-left corner that does not share such x-axis label.

Actioned, see new Fig. 4 Line 253

Figs. 6, S7, S8, S10: The x-axis label is missing.

Added, see new Fig. 5 Line 273, Figs. S9, S10, S14 in Supplemental

Figs. S2-S6: It seems that the legends are raw codes without any explanation in the caption, and it would be difficult to understand for people who's not familiar with these codes.

New Table S1 added with meta data for all chronologies, see Supplemental

L285-287: The readers will wonder why it is the case, and a pointer to the specific discussion section is needed here.

*Northern Hemisphere high altitude/latitude chronologies predominantly used to determine the climatic effects of volcanic eruptions contain higher biological persistence than the chronologies we have used. In terms of the discussion in this paper, we would classify these trees as extreme stress tolerators. To illustrate, we refer to Table 9 in Cropper & Fritts (1981) which compares the ring width characteristics of arctic trees to more temperate Northern Hemisphere trees. The average first order autocorrelation of the series used to develop our temperature reconstructions is 0.53 (0.149-0.869) with standard deviation 0.15 compared to artic sites with average 0.62 (0.15 – 0.93) and standard deviation of 0.13.*

Explanation added at Lines 516-524

L382: It seems that Figure 3 is the one to refer to, instead of Figure 2.

N/A – Figure 3 removed from manuscript.

L458-460: A pointer to Figure 7 is needed here.

Added Line 472

L478: "MDX" is a typo.

Addressed, Line 511

**Detailed response to Reviewer 2**

**General Comments**

Higgins et al. show that New Zealand tree rings can indeed record past volcanic events. They effectively address the research questions they set out to answer. They find that the nature of the response to volcanic dimming varies across species, categorizing species as either "fast responders" of "stress tolerant." With this mixed response between species, they find that site-related factors are more important to the displayed volcanic response in tree-ring width. They additional develop two austral summer temperature reconstructions for New Zealand, which show evidence of cooling from past volcanic events. The response to past volcanic eruptions in these reconstructions shows good agreement with climate model temperature anomalies following volcanic eruptions. The authors competently shows that New Zealand tree-ring width is a reliable regional indicator of volcanic climate response. They add further nuance however and underline the importance of species/site selection, which will be very useful for future studies in this region that wish to optimize sample selection. I believe this publication is fit for publication after minor revisions and will be useful to the research community.

**Specific Comments**

In general I think you need to be more specific with the use of the term "dimming". I'm assuming you're using this term to refer to the increase in SAOD but this should be clearly stated to avoid confusion. The "dimming" term is used throughout the text as a catch all for the effects that could affect tree-ring width, but add specificity where you can. There also needs to be more discussion on the effect of light availability changes, or dimming, and how it could effect final tree-ring width. Particularly in your discussion of the kauri growth benefit (line 393-394). Line 62-63 is another part of the text with opportunity to add more discussion on effects of radiation changes from volcanic eruptions. Here are some references you could use to expand this discussion:

Robock, A. (2005). Cooling following large volcanic eruptions corrected for the effect of diffuse radiation on tree rings. Geophysical Research Letters, 32(6). https://doi.org/10.1029/2004gl022116
Tingley, M. P., Stine, A. R., & Huybers, P. (2014). Temperature reconstructions from tree- ring densities overestimate volcanic cooling. Geophysical Research Letters, 41(22), 7838– 7845.

*In terms of dimming, text has been clarified wherever possible whether we are referring specifically to the increase in SAOD or to cooler temperatures because of dimming.*

Text clarified Lines 21, 68, 324, 343

*Our assumption that kauri receives a growth benefit from decreased evaporative demand following volcanic events is due to previous studies using dendrometer bands and the results of this study. Fowler et al. (2005) show that kauri growth rates are greatest over the austral spring (Sept-Nov), declining steeply over the summer months when evapotranspiration exceeds precipitation. Their results are not entirely consistent with an earlier study by Palmer & Ogden (1983), which showed peak growth continuing until the mid-summer before declining steeply. However, the sites included in Palmer & Ogden were at a higher altitude (245–720 m) than the site in Fowler et al., which could explain the delay in timing. Critical to our moisture stress hypothesis, Palmer & Ogden did not see a summer cessation of growth in their highest altitude site, Mt Moehau, which receives moisture from condensation and fog drip as well as rainfall. This additional information should be included in the discussion to support our assertions. Nevertheless, further discussion of the potential benefit of light availability changes has also been included.*

Discussion added at Lines 62-63, 361-364, 396-403, 454-457 with reference to Tingley et al., (2014); Gu et al., (2003); Zweifel et al., (2021); Fatichi et al., (2019).

Line 109-114 How robust is this event list? Is there a secondary dataset you could use to test? Would you get the same events with the same SAOD thresholds? If there isn't a comparable dataset, I'm not too concerned with this, but I think the choice of this dataset over potential others needs to be explained if it can change the final list of events used.

*Yes, other datasets exist, for example, Crowley & Unterman (2012) and Gao et al. (2008). We used Toohey & Sigl (2017) as it is the most recent compilation of ice core data and has been used in other tree-ring studies of*

*volcanic impacts, e.g., (Rao et al., 2019; Zhu et al., 2020). In response, we have compare the event selection from our study with the SAOD estimates of Crowley & Unterman. Crowley & Unterman provide their estimates of SAOD in two latitudinal bands for the Southern Hemisphere (0-30°S and 30-90°S), and thus we cannot select the same regional threshold (30-50°S) used in the main paper. Instead, we have chosen the Southern Hemisphere average across the two bands, as this was the most consistent with our original threshold. Event selection between the two datasets is largely consistent as are the SEA results, showing a significant t+1 response in both NZall and NZsens temperature reconstructions. Potential reasons for the differences, including the underlying ice core data and differences in methodology, are discussed in Toohey & Sigl (2017).*

SEA analysis of the temperature reconstructions repeated with the dataset from Crowley & Unterman (2013). Results added in Table S3 and Fig. S12 in Supplemental. Discussion of results added to main manuscript at Lines 482-489.

Line 438-439 You need to support the statement that sites with high exposure to prevailing winds are more sensitive to low growing season temperatures, either from the literature or from your own analysis.

*These points are made in reference to the conditions at the individual kauri sites, which has been made clearer with reference to Dang et al. (2007) and Rozas et al. (2013)..*

Additional analysis added at Lines 458-467.

Line 444 Similar to the point above, you need to support this statement.

*However, evidence to support the lower sensitivity of sites experiencing mesic conditions and closed-canopy forests has been added with reference to Phipps (1982).*

Lines 453-454.

Figure 1 Add a legend for the elevation. This is important context for your conclusions as elevation is an important site characteristic.

Added, revised Fig. 1 Line 107.

Figure 3 Caption "...the number of chronologies are shown in brackets/square brackets." Make it clear which bracket type refers to which chronology. Adding the word "respectively" will work.

Clarified, see Line 201.

**Technical**

Line 22 proxy --> site/species Using proxy sounds like you are expanding into non tree- ring proxies like coral for example.

Changed to 'chronology' Line 24

Line 46-49 Awkward sentence structure

Text changed Line 51-54

*There are several potential explanations for the considerable discrepancy between proxy reconstructions and climate models in the Southern Hemisphere. These include the underestimation of the moderating effects of the ocean on post-eruption cooling in climate models, changes to the hydrological cycle in response to volcanic cooling, uncertainties in volcanic forcing data, and/or proxy noise and spatial distribution (Neukom et al., 2018; Zhu et al., 2020).*

Line 51-52 tree-ring data

Changed Line 58

Line 89 proxy --> site/species

Changed to 'chronology' Line 96

Line 114-115 awkward sentence structure

Text changed Line 132-134

*The SAOD magnitude corresponding to a substantial temperature response is unknown before analysis. However, selecting a magnitude post-analysis based on the observed response risks biasing the results (Haurwitz and Brier, 1981). Therefore, two different SAOD thresholds were used...*

Line 140 specify season

DJF specified Line 156.

Line 436 add a call to Figure 5

Added Line 441.

Line 458-460 awkward sentence structure

Text changed Line 470-472.

*We expected to find a substantially greater volcanic response in NZsens (i.e., limited to only those chronologies with an individual significant volcanic response), compared to NZall. However, while NZsens does show a larger post-volcanic temperature response the difference between the two reconstructions is not significant (Fig. 6).*

Line 478 MDX-->MXD typo

Changed, Line 511.

Line 482 Add call to Figure 7

Added Line 515.

Line 508 proxy --> species/site

Changed to chronology, Line 529.

**Detailed response to Community Comment 1**

**General comments**

This paper led by Higgins examines the ability of eight New Zealand tree species to reflect volcanic dimming. As the authors point out, little is known of the impacts of past volcanic eruptions on Southern Hemisphere climate, and there is a discrepancy between models and palaeo-data. The authors found that there are differences across and within species in terms of their apparent reaction to dimming. The authors also present a summer temperature reconstruction from the tree-rings that reflects influence of dimming. This same response is detected in the 7-model ensemble response. Previous studies on Southern Hemisphere trees have not identified a volcanic impact. This study is therefore of considerable interest and importance. Overall, the manuscript is quite well written, although some work is required to improve clarity and succinctness, and to better emphasise main findings.

I wonder if a rearrangement of the material would help emphasise the key findings of this study a little better, better follow on from the introduction and also provide a basis for stronger and more substantial conclusions. One suggestion might be to start with showing a response in a new [more fully described] temperature reconstruction (or reconstructions) composed of a multi-species network of temperature sensitive chronologies to regional volcanic dimming, and a comparison with the CMIP ensemble before delving into the details of species used and species-level responses? The reason I suggest this, is that the introduction (l. 40 – 49, 65) seems to indicate that the model-paleo discrepancy will be an important aspect of the paper, but this doesn't really come through (might the choice of a regional dimming index be relevant? – see below). I think that finding a volcanic response is the first 'big' result of the study. I am guessing the authors view it as more useful to build the case from the sites/species first and then to look at the temperature reconstruction but nevertheless ask that they carefully consider what aspects of their work deserve greatest emphasis.

*We believe there are two 'big' results from this study, and both deserve to be highlighted in the paper.*

> *a) opposing volcanic signals can be identified in chronologies from the same species, and*

> *b) there is a clear Southern Hemisphere volcanic dimming signal.*

*While the first result may be predominantly of interest to dendrochronologists and the second to the wider paleoclimate community, we don't believe either is of greater importance. As acknowledged, the paper is structured to build a case for temperature reconstructions from the site/species information. It is also structured this way to highlight both significant results.*

Not actioned. We believe the paper structure is appropriate in its current form.

The authors' use of a regional dimming index relevant to New Zealand latitudes/longitudes rather than a selection of events based on eruption magnitude may be a key reason for their results. I think this needs to be discussed in more detail – it seems to be the elephant in the room when the authors are focussed on reasons for differences in responses amongst species. The fact that a relationship with volcanic eruptions has been identified in temperature-sensitive Southern Hemisphere trees is highly newsworthy. Why/how did the authors find this when other studies haven't? It would be good to place considerably more emphasis on this in the Discussion/Conclusions.

A comparison to previous studies with explanation of the difference to our study added at Lines 490-501.

The authors rightly mention the moderate temperature response of the New Zealand species. This also applies to the ring widths of other Southern Hemisphere species. Do the authors consider that their use of the regional dimming index might 'compensate' somewhat for this moderate temperature response?

*Using a regional dimming index to select events rather than an eruption magnitude almost certainly has contributed to identifying volcanic signals in this study. This was the intention of the event selection method, and we believe it was the right choice for this study. However, we don't think of the selection method as 'compensating' for a moderate temperature response. All tree ring-volcano studies select a threshold for which events to include, and this is known to be a significant source of uncertainty. Whichever the chosen threshold, studies aim to select events that would have resulted in a climate response in their study region/hemisphere. We used two different*

*event thresholds and bootstrapped confidence intervals to account for this uncertainty in our results. The 90th percentile bootstrapped confidence intervals show that event selection could substantially impact the observed temperature response.*

SEA analysis of the temperature reconstructions repeated with the dataset from Crowley & Unterman (2013). Results added in Table S3 and Fig. S12 in Supplemental. Discussion of results added to main manuscript at Lines 482-489.

In relation to the more detailed analysis of species and sites, it would also be useful to show more information on the actual sites. It is not really sufficient to state that the meta-data for sites can be found in Palmer et al. 2015. It is difficult to adequately consider some of the points made by the authors in the discussion without having the sites put into context much earlier. For example, the information in Table 2 (along with references to the supplementary Figure 2) could be presented in Section 2.1. A summary (possibly pictorial and perhaps in the Supplementary?) of the various species' sites by altitude/location would also be very helpful to better guide the reader through the results/discussion. Would such a figure help when providing some detail on which sites within a species did not have a strong volcanic response (Section 3.1)? Are there common factors – like altitude for example – that play a role in nonsignificant response within species? Could it be linked in any way to the reason for previous studies not finding a relationship with volcanic eruptions? (e.g. the authors discuss elevation and latitude).

New Table S1 with all site meta data added to Supplementary. A summary of the temperature sensitivity of the different species added to Section 2.1 with reference to the supplementary figures.

Also, the authors comment in the conclusions that only a subset of the temperature sensitive chronologies show a response to volcanic eruptions. Figures S2-6 show that a number of the chronologies that are not temperature sensitive.

    a)   If the argument is that volcanic eruptions affect temperature and it is this that then impacts radial growth, why not use this information to exclude chronologies from the reconstruction and/or the analysis for a volcanic signal in the first place?

        *While there are broadly consistent temperature sensitivities for each species group, as the reviewer highlights, different chronologies show different strengths in these relationships (Figures S2-S6 – please note only chronologies extending until 1990, and thus considered for the temperature reconstructions, were plotted on these figures). Considering the full suite of 96 chronologies in this study, only four (1HUI.r, 1MOE.r, 1MWL.r, 8WKT.r) shows no significant ($p < 0.05$) correlations to NZ average temperature in any month over the two growing seasons. As this is the likely criterion to be used to exclude chronologies from volcanic analysis, the change to the results would be negligible.*

        Not actioned.

    b)   Some better links between this information and discussion of which sites do and do not show a volcanic response (or show a range of responses – i.e. cedar) may be warranted. Ditto in terms of positive/negative responses – does a dominant current season [positive] temperature response equate to a negative response to volcanic eruptions? Does a dominant prior season [negative] temperature response equate to a positive response to volcanic eruptions? (for eg)?

        Considerable reworking of text to highlight these links at Line 218-232, and Line 421-428. Addition of new Fig. S15 in Supplementary.

    c)   Any comment on seasonal window of temperature response and its relevance (or not) to volcanic response?

        *The seasonal sensitivity of tree growth is potentially an important determinant of the volcanic response, but we don't have sufficient evidence from the results of this study to support a discussion. It would be very interesting to investigate whether the whole-of-season sensitivity of pink pine causes more or less sensitivity to climate disturbance compared to the other species with narrower temperature sensitivity, but this is a question for future research. A greater number of (updated) sites with two or more species*

*(i.e., pink pine and cedar) would allow a comparison of the role of seasonal sensitivity largely without other confounding factors.*

Discussion added at Line 371-373.

Loosely linked to this point, the authors make the case that lower elevation sites have a temperature response related to moisture stress (l. 432-434, 503). While this is not an unreasonable suggestion, the authors need to be more careful about how they state this – they have not shown data to support the statement, so comments should be more cautious when it is mentioned.

*Yes, the temperature-moisture stress response at low elevation sites is a hypothesis and should be qualified as such.*

Additional explanation added and language clarified Lines 434-440.

At face value, there seems to be an inherent contradiction in the authors' discussion around 'stress tolerators' and delayed responses and their later comments about the lack of a lagged response to volcanic eruptions in the temperature reconstruction. It is important to clarify this given that the memory in tree-rings has been found to be an issue in the response of Northern Hemisphere trees to volcanic eruptions. The authors should carefully consider what they are implying in their discussion of lagged response (as shown in Figure 3 and S1) as opposed to their comments about the 'lack of memory' in the temperature reconstruction. Why are there apparently lagged responses of varying magnitude across the chronologies that are not apparently reflected in the reconstruction? Some careful consideration needs to go into this.

*Firstly, the suite of tree rings used in this study have lower biological persistence compared to published Northern Hemisphere ring-width records (Esper et al., 2015). 80% of the predictors used to develop the NZall reconstruction only have significant lag-1, or lag-1 and lag-2 autocorrelation after standardisation. So, while we see lagged effects after eruptions in the composite for the stress tolerators due to persistence, there is less lag compared to arctic trees.*

*Reconstruction methodology also plays an important role in the persistence of the final temperature reconstructions compared to the predictors used. We refer to the study of Büntgen et al. (2021), which compares 15 temperature reconstructions developed by different research groups with different methodologies but using the same set (or a subset) of chronologies. AR1 persistence in the final reconstructions varied from < 0.4 to > 0.9 due to methodological decisions, and there was a substantial variation in response to volcanic cooling between the reconstructions (Figure S2, their study). The important elements of our reconstruction methodology which likely contribute to the final autocorrelation structure are a) pre-whitening of both the tree rings and temperature data prior to reconstruction and b) testing predictors for significance in both year t and t+1.*

Discussion added Lines 516-524.

Why were the specific 7 CMIP models selected (Table S2)? Why not other models?

*These were the models that could be freely downloaded from the ESGF@DOE/LLNL, and met the condition of having both a past1000 and historical run.*

There is quite a bit of repetition (almost the same sentences in some cases) between the Results and Discussion. This should be minimised, especially given that the authors are presenting a range of interesting results across and within species groups, and a temperature reconstruction and its response that is compared with a model ensemble. It is important to draw the threads together as coherently as possible.

Considerable reworking of Results and Discussion sections has taken place.

A minor point: the title implies Southern hemisphere, but the study is focused on NZ. Perhaps it would be pertinent to include "A case study from New Zealand" or similar in the title.

Title changed to '*Do Southern Hemisphere tree rings record past volcanic events? A case study from New Zealand*'

***Specific comments***

**Introduction**

1.  47-9 Might seasonality of signal matter?

    Discussion added at Line 371-373.

2.  52-5 "Tree growth…." Yes, but this almost sounds like the vast majority of the SH trees should be ruled out simply based on location and lack of serious elevation. It also seems to differ from what some of the results suggest (low elevation sites in mid-latitudes apparently sensitive).

    *This is not the intention of the paragraph. We wish to highlight that it may be harder to identify volcanic signals from less temperature–limited trees, as indicated by similar studies in the Northern Hemisphere.*

    Not actioned.

3.  66 – 7 – This sentence doesn't follow previous. Why would understanding whether a site is likely to have a volcanic response necessarily be relevant for studies of hydroclimate?

    We are not sure to which sentence the reviewer is referring. Sentence 66-67 reads, *"This knowledge will benefit future studies of hemispheric temperatures and help identify which species and/or regions should be prioritised for future proxy development."*

    Not actioned as comment unclear.

4.  73-74. Reword a little – awkward to read.

    Revised text Line 80-82

    *Land clearing has resulted in the loss of forests from most lowland areas and nearly all of the eastern drylands. The most common remaining forest types are wet conifer-broadleaved forests and montane to alpine southern beech (Nothofagaceae) dominated forests (McGlone et al., 2017).*

**Methods**

5.  104 maybe reword this first sentence slightly to improve clarity.

    Revised text Line 124

    *Event selection is a significant source of uncertainty in tree-ring studies of volcanic cooling.*

6.  113 10 and 18 eruptions. On next page on l. 126 and also l. 118, 13 and 21 events? Seems to be an error here? In any case, this is confusing.

    *10 and 18 refer to the number of events between 1400 and 1900 CE, including the three events subsequently happening in the 1900s, bringing the event lists to 13 and 21.*

    Text clarified Line 138-139.

7.  l.116 "Between 1900 and 1990, we selected the three largest.." Where these the largest based on the same criteria for selecting the historical eruptions? (ie. based on the regional dimming index?)

*The dataset we used for the latitudinally modelled SAOD (Toohey and Sigl, 2017) doesn't extend past 1900. Therefore, event selection from 1900 was not based on the same dimming index. Please refer to the response to RC2 to see the impact on the results from using a different dataset to set the thresholds.*

8. 128 "Species-level…". This again makes me wonder if it would be wise to first screen out those sites for each species that do not have a strong temperature response?

Not actioned as would have negligible impact on the results – see full response above.

9. l.153 DJF Maybe point to Table 1 as justification – but include actual months of sensitivity in Table 1 – see below for comment.

Seasonal sensitivities for all species added to Table 1 and Table 1 referenced Line 164.

10. 165 – 169. Presumably for DJF so comparable with the tree-ring reconstruction

Clarified Line 186.

**Results**

11. 173 – 175 rewrite this a bit to be as clear as possible.

Text revised Line 193-196.

*The results of the superposed epoch analysis for the 13 largest volcanic eruptions between 1400 and 1990 CE are shown in Figure 3. Two composite responses are shown for each species; the response averaged across all sites ('All chronology composite'), and the response calculated only from the site chronologies which individually showed a significant (either positive or negative) response to volcanic eruptions ('Sensitive chronology composite').*

12. 178 Which species had a neutral response? List here.

Added Line 207.

13. 180 – 195 Include references to lagged responses shown in Figure…compare with later comments on the 'lack of memory' in the NZ trees compared to the NH trees. Maybe also better link this with the nature of the temperature responses (Figure S2-6).

Discussion added Lines 520-524.

**Section 3.2**

14. I am tempted to suggest that this could potentially go in Supplementary material (or even be omitted altogether) to simplify the paper and amplify the main points of the paper. I think it is useful to look at this, but not a key point. Also, the discussion here is a little confusing. In some cases it is the difference between the species at the various sites that is noted but not really described fully, but in other places, both species record a negative response. It would be useful to discuss both the nature of the responses of the individual species at these sites and then if they differ from the other species.

Section removed. Fig. 4 moved to Supplemental as Fig. S8.

**Section 3.3**

15. 218 It would be good to preface this section with some statement about why focus on cedar (why not do similar analyses for other species? – i.e. be explicit). One suggestion…begin with a comment about how the cedar average showed a generally muted response, but this masks very different individual site responses…and hence why this section of the results is useful. While it is certainly understandable that the authors wish to consider this material in the main manuscript, it may be worth considering whether

at least some of this information could go in the Supplementary? (so as not to distract too much from the bigger messages in the paper).

*With Section 3.2 moved to Supplementary, we think the results of Section 3.3 are important to be included in the main paper.*

Added explanatory text Line 234-236.

16. 279 models – not CMIP models? Just be clear which models (reconstruction model or CMIP) is being referred to.

    Clarified Line 292.

17. 284 – 287 So there isn't a substantial issue with memory in the temperature reconstruction, but there are lags in the species-level responses. Be careful how this is discussed throughout the Results and Discussion.

    Addressed Lines 520-524.

**Discussion**

18. As mentioned earlier, there is considerable repetition here (representation of Results) that clouds the text somewhat.

    Considerable reworking of Discussion with repetitive text deleted.

19. Section 4.1 – Compare with the above. Why doesn't this play out in the temperature reconstruction? Would be good to comment on.

    Addressed Lines 520-524.

20. 340 – 345 This seems a little confused. Why separate silver beech from the other stress tolerators in this section? This section could probably be tightened up a bit.

    *In this paragraph, we compared the responses of the two beech species, as distinct from the remaining species, which are all conifers. We found it interesting that the beech species showed such different response characteristics. However, this discussion has been incorporated into the previous paragraphs to make it clearer for the reader.*

    Paragraph reordered Line 333-352.

21. 350 – 354. So how does this relate to strong responses in years 0 and 3?

    *We propose that the initial response in year 0 is foliage production, and in year 3, a secondary response is triggered by the cooler than average summer temperatures in year 0. Whether this response is due to a quasi-biennial flowering cycle or mast seeding or is related to normal cladode lifespan is a hypothesis. We favour the cladode senesce hypothesis due to our own, albeit limited, field observations.*

    Explanation revised Line 353-364.

22. 369 "…around 1000mm" This still seems relatively high, but how does it relate to the needs and distributional range (with respect to precipitation) of the species?

    Due to revisions including the removal of Section 2.3, this discussion was deleted.

23. 370 – 373. Again, mention of possible link to role of moisture. This seems to suggest that perhaps moisture-related variability should have been considered in this study? However, the low number of samples is of some concern, and perhaps this section should be shortened accordingly.

Due to revisions including the removal of Section 2.3, this discussion was deleted.

24. 392 I'm not sure this is counter-intuitive response given the negative response to temperature (Supplementary). L. 397-99. This mention of seasonality of growth again makes me wonder if this should have been more fully described for all species much earlier (the longer climate response window of pink pine for eg is interesting)?

Wording revised Line 387. Summary of climate sensitivities added to Section 2.1 and Table 1.

25. Section 4.2 Needs considerable tightening up. Reference to moisture-related responses is speculative (but not unreasonable), but it needs to be couched that way. Also, while the results and discussion for cedar in Section 3.3 are suggestive, I don't think they should be presented as being THE causes of differences. They may well be, but further work, and closer examination across all the species would provide more evidence for this. At l. 433, other factors are mentioned. This reference should perhaps come earlier in this section to better set up the discussion around the evidence presented in Section 3.3. Especially when the authors go on to discuss location in the landscape (l. 440 – 446). This isn't discussed in relation to the PCA results in Section 3.3. If these factors are so important, they should be mentioned in that section – do the PCA results reflect this?

Considerable reworking of Discussion to address multiple comments resulted in Section 4.2 being deleted. Some text from Section 4.2 has been incorporated into Section 4.1 at Lines 365-379.

26. 422 " …sensitivity to temperature, including volcanic cooling…" So why include sites/chronologies not sensitive to temperature (or not at their limits)? This gets back to the locations of many of the SH tree-ring chronologies, and the relative lack of 'choice' compared to the NH.

Discussion added at Line 422-428.

27. 450 "..large on site-related…". Only Elevation and latitude really mentioned.

Considerable reworking of Section 4.3 to discuss other site-related factors.

28. 450 – 452. While it is great that the authors produced this new reconstruction and tested it for volcanic response, I have two points to make about it:

   a. The main features of this (these) new reconstruction are not really described in the study. How does it differ from earlier reconstructions? (Obviously the climate target may differ, but does it show similar features? If not/if so, where….?). Is it different enough to be the potential reason for volcanic dimming being detectable here but not in previous reconstructions?

   Discussion of key features and comparison to previous studies added at Line 279-285 and addition of Fig. S11 and Table S5 to Supplemental.

   b. If previous reconstructions were compared with the regional dimming index in the same manner would the same result be produced? Has it been the compilation of volcanic eruptions based on their magnitude which has been the problem in the past? Or is it the combination of chronologies used? The target season? The season of the eruptions vs seasonality of tree growth without due attention to regional and global circulation patterns?

   *None of the studies discussed above considered a volcanic response, nor did any of the other published NZ temperature reconstructions for which the data are not available. Thus, we suspect the main reason volcanic responses have not been previously identified in New Zealand tree-ring temperature reconstructions is that no analysis has been done until now.*

   *Palmer & Ogden (1992) and Norton (1992) are the only studies to our knowledge that looked for volcanic signals in tree rings from New Zealand. Both studies consider only the Tambora eruption of 1815, and their methodologies simply identify whether there were narrow rings in the years following the eruption event. Both conclude that the evidence is not sufficient to identify a volcanic signal. Our analysis and the results of Palmer et al. compare quite well: cedar chronologies declined for several years following an eruption, similar to the lagged and persistent pattern we observed over 13 eruptions, and the two years following the*

*eruption show increased average ring width in their kauri chronologies. Similarly, our results for the two beech species correspond well with the results of Norton. The issue faced by these researchers is that they were seeking common responses across species, whereas our analysis, with the benefit of much more data, shows responses vary widely between species.*

*Without undertaking a detailed analysis, the global investigation of Krakauer & Randerson (2003) likely suffers from the compositing of the many different chronologies, which results in some of the volcanic signal cancelling as noise. This is the reason we caution against compositing too widely in our conclusions.*

Explanation added Line 490-501.

**Conclusions**

29. I still think the big news is that a volcanic signal was identified. The 'next big news' is related to the species-level responses.

   Not actioned. We believe the structure is appropriate in its current form.

30. 488 "..proxy selection…". This almost sounds like a choice amongst corals, trees, speleothems etc. Do you mean site selection?

   Change to 'chronology' Line 550.

31. 492 "….plant life history traits…". What is meant by this? Not really discussed in the manuscript. Maybe just be explicit to avoid confusion.

   Revised discussion is Section 4.1 should clarify this point.

32. 500 "We found that…." Not convinced this is a major finding when it depends heavily on Section 3 and then later observations not related to the analysis in Section 3.3.

   Text has been revised Line 539-547.

33. 503 " …summer moisture…" I think this is too speculative to include in the conclusions.

   Text has been revised Line 539-547.

34. 513-515. But this study seemed to indicate that it didn't matter so much whether a subset of the most sensitive sites was used or not (especially for the temperature reconstruction). Reword.

   Text removed.

35. 518 – 521. Yes, this applies to other types of reconstructions as well. Note that several large databases include composites that are then used by modellers who may not appreciate these types of nuances.

   No action required.

**Figures and Tables**

36. Figure 2. The orange and red are quite close to one another.  Might it be useful to darken the red so there is a clearer visual difference between the two?

   Colours adjusted in revised Fig. 2.

37. Figure 5 – colours for G4 and G5 difficult to tell apart for some.  Change one of the colours.
   Colours adjusted in revised Fig. 4.

38. Figure 8 – is it possible to use different symbols for the different species?
   Symbols adjusted in revised Fig. 7.

39. Table 1 Could this table be merged with Table 2 that simply summarises the nature of the response. Maybe also note which response is stronger, prior or current season?

Tables 1 and 2 merged. Additional references to Table 1 and Figs. S1-S6 added at Line 111-122.Yes, the Table 1 column 'reported climate sensitivity', which summaries the referenced publications, could be replaced with the calculated responses from Table 2.

40. Table 2 – I think this could be merged with Table 1, but in discussing Table careful references to Figures S2-6 should be made.
Tables 1 and 2 merged. Additional references to Table 1 and Figs. S1-S6 added at Line 111-122.Yes, the Table 1 column 'reported climate sensitivity', which summaries the referenced publications, could be replaced with the calculated responses from Table 2.

**Supplementary**

41. Figures S3 and S4. It is unclear why master chronologies are included just for these two species. Perhaps more could be made of the differences between 'master series' and individual series for all species in the main text? It would actually be good to see master series for all species included here given that species wide averages have been used in the main manuscript (Figure 3).

Considerable reworking of Figs. S1-S6 to address this comment, plus addition of Table S1 in Supplementary.

*Technical comments*

Abstract

42. 28 "The has…" This has.
Addressed Line 31.

Introduction

43. 88 amongst rather than between
Addressed Line 95.

44. 89 Should "proxy" be "site"?
Changed to 'chronology' Line 96.

Methods

45. 98 should "species depth" be "sample depth"?

Text changed Line 105-106

*As only a single chronology has been developed from mountain toatoa (Phyllocladus alpinus) it was excluded from the study*

46. 133 "…two species.." Maybe insert 'different' between these words?
Due to revisions, text has been deleted.

Results

47. 175 "averaged across…." All sites of a species, not just all sites. Ditto in relation to sensitive chronologies.

Text revised Line 193-199.

Discussion

48. 438 "…altitudinal range.." Altitudinal limit?

Text revised Line 450.

49. 478 MDX – should be MXD?

Text revised Line 511.